# Guiding Neural Collapse: Optimising Towards the Nearest Simplex Equiangular Tight Frame

**Evan Markou**
Australian National University
`evan.markou@anu.edu.au`

**Thalaiyasingam Ajanthan**
Australian National University & Amazon
`thalaiyasingam.ajanthan@anu.edu.au`

**Stephen Gould**
Australian National University
`stephen.gould@anu.edu.au`

## Abstract

Neural Collapse (NC) is a recently observed phenomenon in neural networks that characterises the solution space of the final classifier layer when trained until zero training loss. Specifically, NC suggests that the final classifier layer converges to a Simplex Equiangular Tight Frame (ETF), which maximally separates the weights corresponding to each class. By duality, the penultimate layer feature means also converge to the same simplex ETF. Since this simple symmetric structure is optimal, our idea is to utilise this property to improve convergence speed. Specifically, we introduce the notion of *nearest simplex ETF geometry* for the penultimate layer features at any given training iteration, by formulating it as a Riemannian optimisation. Then, at each iteration, the classifier weights are implicitly set to the nearest simplex ETF by solving this inner-optimisation, which is encapsulated within a declarative node to allow backpropagation. Our experiments on synthetic and real-world architectures for classification tasks demonstrate that our approach accelerates convergence and enhances training stability[1].

## 1 Introduction

While modern deep neural networks (DNNs) have demonstrated remarkable success in solving diverse machine learning problems [22, 34, 38], the fundamental mechanisms underlying their training process remain elusive. In recent years, considerable research efforts have focused on delineating the optimisation trajectory and characterising the solution space resulting from the optimisation process in training neural networks [72, 17, 49, 41]. One such finding is that gradient descent algorithms, when combined with certain loss functions, introduce an implicit bias that often favours max-margin solutions, influencing the learned representations and decision boundaries. [44, 57, 33, 27, 21, 54, 70, 31, 49].

In this vein, Neural Collapse (NC) is a recently observed phenomenon in neural networks that characterises the solution space of the final classifier layer in both balanced [50, 76, 74, 28, 48, 63, 43, 32] and imbalanced dataset settings [19, 59, 5]. Specifically, NC suggests that the final classifier layer converges to a Simplex Equiangular Tight Frame (ETF), which maximally separates the weights corresponding to each class, and by duality, the penultimate layer feature means converge to the classifier weights, *i.e.*, to the simplex ETF (formal definitions are provided in Appendix A). This simple, symmetric structure is shown to be the only set of optimal solutions for a variety of loss functions when the features are also assumed to be free parameters, *i.e.*, Unconstrained Feature

---

[1]Code available at https://github.com/evanmarkou/Guiding-Neural-Collapse.git.

38th Conference on Neural Information Processing Systems (NeurIPS 2024).

Models (UFMs) [32, 19, 74, 76, 28, 75]. Nevertheless, even in realistic large-scale deep networks, this phenomenon is observed when trained to convergence, even after attaining zero training error.

Since we can characterise the optimal solution space for the classifier layer, a natural extension is to *leverage the simplex ETF structure of the classifier weights to improve training*. To this end, researchers have tried fixing the classifier weights to a canonical simplex ETF, effectively reducing the number of trainable parameters [76]. However, in practice, this approach does not improve the convergence speed as the backbone network still needs to do the heavy lifting of matching feature means to the chosen fixed simplex ETF.

In this work, we introduce a mechanism for finding the nearest simplex ETF to the features at any given training iteration. Specifically, the nearest simplex ETF is determined by solving a Riemannian optimisation problem. Therefore, our classifier weights are dynamically updated based on the penultimate layer feature means at each iteration, *i.e.*, implicitly defined rather than trained using gradient descent. Additionally, by constructing this inner-optimisation problem as a deep declarative node [23], we allow gradients to propagate through the Riemannian optimisation facilitating end-to-end learning. Our whole framework significantly speeds up convergence to a NC solution compared to the fixed simplex ETF and conventional learnable classifier approaches. We demonstrate the effectiveness of our approach on synthetic UFMs and standard image classification experiments.

Our main contributions are as follows:

1. We introduce the notion of the nearest simplex ETF geometry given the penultimate layer features. Instead of selecting a predetermined simplex ETF (canonical or random), we implicitly fix the classifier as the solution to a Riemannian optimisation problem.
2. To establish end-to-end learning, we encapsulate the Riemannian optimisation problem of determining the nearest simplex ETF geometry within a declarative node. This allows for efficient backpropagation throughout the network.
3. We demonstrate that our method achieves an optimal neural collapse solution more rapidly compared to fixed simplex ETF methods or conventional training approaches, where a learned linear classifier is employed. Additionally, our method ensures training stability by markedly reducing variance in network performance.

## 2   Related Work

**Neural Collapse and Simplex ETFs.**   Zhu et al. [76] proposed fixing classifier weights to a simplex ETF, reducing parameters while maintaining performance. Simplex ETFs effectively tackle imbalanced learning, as demonstrated by Yang et al. [67], where they fix the target classifier to an arbitrary simplex ETF, relying on the network's over-parameterisation to adapt. Similarly, Yang et al. [68] addressed class incremental learning by fixing the target classifier to a simplex ETF. They advocate adjusting prototype means towards the simplex ETF using a convex combination, smoothly guiding backbone features into the targeted simplex ETF. However, these methods did not yield any benefits regarding convergence speed. The work most relevant to ours is that of Peifeng et al. [51], who argued about the significance of feature directions, particularly in long-tailed learning scenarios. They compared their method against a fixed simplex ETF target, formulating their problem to enable the network to learn feature direction through a rotation matrix. Additionally, they efficiently addressed their optimisation using trivialisation techniques [39, 40]. However, they did not demonstrate any improvements in convergence speed over the fixed simplex ETF, achieving only a minimal increase in test accuracy. Fixing a classifier is not a recent concept, as it has been proposed prior to the emergence of neural collapse [52, 58, 30]. Most notably, Pernici et al. [52] demonstrated improved convergence speed by fixing the classifier to a simplex structure only on ImageNet while maintaining comparable performance on smaller-scale datasets. In contrast, our method shows superior convergence speed compared to both a fixed simplex ETF and a learned classifier across both small and large-scale datasets.

**Optimisation on Smooth Manifolds.**   Our optimisation problem involves orthogonality constraints, characterised by the Stiefel manifold [2, 11]. Due to the nonlinearity of these constraints, efficiently solving such problems requires leveraging Riemannian geometry [18]. A multitude of works are dedicated to solving such problems by either transforming existing classical optimisation techniques into Riemannian equivalent algorithms [1, 73, 20, 64, 55] or by carefully designing penalty functions

to address equivalent unconstrained problems [66, 65]. In our approach, we opt for a retraction-based Riemannian optimisation algorithm [1] to optimally handle orthogonality constraints.

**Implicit Differentiable Optimisation.** In a neural network setting, end-to-end architectures are commonplace. To backpropagate solutions to optimisation problems, we rely on machinery from implicit differentiation. Pioneering works [4, 3] demonstrated efficient gradient backpropagation when dealing with solutions of convex optimisation problems. This concept was independently introduced as a generalised version by Gould et al. [23, 24] to encompass any twice-differentiable optimisation problem. A key advantage of Deep Declarative Networks (DDNs) lies in their ability to efficiently solve problems at any scale by leveraging the problem's underlying structure [25]. Our setting involves utilising an equality-constrained declarative node to efficiently backpropagate through the network.

## 3 Optimising Towards the Nearest Simplex ETF

In this section, we introduce our method to determine the nearest simplex ETF geometry and detail how we can dynamically steer the training algorithm to converge towards this particular solution.

### 3.1 Problem Setup

Let us first introduce key notation that will be useful when formulating our optimisation problem.

**Simplex ETF.** Mathematically, a general simplex ETF is a collection of points in $\mathbb{R}^C$ specified by the columns of a matrix

$$\boldsymbol{M} = \alpha\sqrt{\frac{C}{C-1}}\boldsymbol{U}\left(\boldsymbol{I}_C - \frac{1}{C}\boldsymbol{1}_C\boldsymbol{1}_C^\top\right) . \tag{1}$$

Here, $\alpha \in \mathbb{R}_+$ denotes an arbitrary scale factor, $\boldsymbol{1}_C$ is the $C$-dimensional vector of ones, and $\boldsymbol{U} \in \mathbb{R}^{d \times C}$ (with $d \geq C$) represents a semi-orthogonal matrix ($\boldsymbol{U}^\top\boldsymbol{U} = \boldsymbol{I}_C$). Note that there are many simplex ETFs in $\mathbb{R}^C$ as the rotation $\boldsymbol{U}$ varies, and $\boldsymbol{M}$ is rank-deficient. Additionally, the standard simplex ETF with unit Frobenius norm is defined as: $\tilde{\boldsymbol{M}} = \frac{1}{\sqrt{C-1}}\left(\boldsymbol{I}_C - \frac{1}{C}\boldsymbol{1}_C\boldsymbol{1}_C^\top\right)$.

**Mean of Features.** Consider a classification dataset $\mathcal{D} = \{(\boldsymbol{x}_i, y_i) \mid i = 1, \ldots, N\}$ where the data $\boldsymbol{x}_i \in \mathcal{X}$ and labels $y_i \in \mathcal{Y} = \{1, \ldots, C\}$. Suppose, $n_c$ is the number of samples correspond to label $c$, then $\sum_{c=1}^C n_c = N$. Let us consider a scenario where we have a collection of features defined as,

$$\boldsymbol{H} \triangleq [\boldsymbol{h}_{c,i} : 1 \leq c \leq C, 1 \leq i \leq n_c] \in \mathbb{R}^{d \times N} . \tag{2}$$

Here, each feature may originate from a nonlinear compound mapping of input data through a neural network, denoted as, $\boldsymbol{h}_{y_i,i} = \phi_{\boldsymbol{\theta}}(\boldsymbol{x}_i)$ for the data sample $(\boldsymbol{x}_i, y_i)$. Now, for the final layer, our decision variables (weights and biases) are represented as $\boldsymbol{W} \triangleq [\boldsymbol{w}_1, \ldots, \boldsymbol{w}_C]^\top \in \mathbb{R}^{C \times d}$, and $\boldsymbol{b} \in \mathbb{R}^C$, and the logits for the $i$-th sample is computed as,

$$\psi_{\boldsymbol{\Theta}}(\boldsymbol{x}_i) = \boldsymbol{W}\boldsymbol{h}_{y_i,i} + \boldsymbol{b} , \qquad \text{where} \quad \boldsymbol{h}_{y_i,i} = \phi_{\boldsymbol{\theta}}(\boldsymbol{x}_i) . \tag{3}$$

In UFMs, the features are assumed to be free variables, which serves as a rough approximation for neural networks and helps derive theoretical guarantees. Additionally, we define the global mean and per-class mean of the features $\{\boldsymbol{h}_{c,i}\}$ as:

$$\boldsymbol{h}_G \triangleq \frac{1}{N}\sum_{c=1}^C\sum_{i=1}^{n_c}\boldsymbol{h}_{c,i} , \quad \bar{\boldsymbol{h}}_c \triangleq \frac{1}{n_c}\sum_{i=1}^{n_c}\boldsymbol{h}_{c,i} , \quad (1 \leq c \leq C) , \tag{4}$$

and the globally centred feature mean matrix as,

$$\bar{\boldsymbol{H}} \triangleq [\bar{\boldsymbol{h}}_1 - \boldsymbol{h}_G, \ldots, \bar{\boldsymbol{h}}_C - \boldsymbol{h}_G] \in \mathbb{R}^{d \times C} . \tag{5}$$

Finally, we scale the feature mean matrix to have unit Frobenius norm, *i.e.*, $\tilde{\boldsymbol{H}} = \bar{\boldsymbol{H}}/\|\bar{\boldsymbol{H}}\|_F$ which will be used in formulation below.

## 3.2 Nearest Simplex ETF through Riemannian Optimisation

Once we obtain the feature means, our objective is to calculate the nearest simplex ETF based on these means and subsequently adjust the classifier weights $\boldsymbol{W}$ to align with this particular simplex ETF. The rationale is to identify and establish a simplex ETF that closely corresponds to the feature means at any given iteration. This approach aims to expedite convergence during the training process by providing the algorithm with a starting point that is closer to an optimal solution rather than requiring it to learn a simplex ETF direction or converge towards an arbitrary one.

To find the nearest simplex ETF geometry, we solve the following Riemannian optimisation problem

$$\underset{\boldsymbol{U} \in St_C^d}{\text{minimize}} \left\| \tilde{\boldsymbol{H}} - \boldsymbol{U}\tilde{\boldsymbol{M}} \right\|_F^2 \tag{6}$$

where $St_C^d = \{ \boldsymbol{X} \in \mathbb{R}^{d \times C} : \boldsymbol{X}^\top \boldsymbol{X} = \boldsymbol{I}_C \}$. Here, $\tilde{\boldsymbol{M}}$ is the standard simplex ETF with unit Frobenius norm, and the set of the orthogonality constraints $St_C^d$ forms a compact Stiefel manifold [2, 11] embedded in a Euclidean space.

## 3.3 Proximal Problem

The solution to the Riemannian optimisation problem, denoted as $\boldsymbol{U}^\star$, is not unique since a component of $\boldsymbol{U}^\star$ lies in the null space of $\tilde{\boldsymbol{M}}$. As simplex ETFs reside in $(C-1)$-dimensional space, the matrix $\tilde{\boldsymbol{M}}$ is rank-one deficient. Consequently, we are faced with a family of solutions, leading to challenges in training stability, as we may oscillate between multiple simplex ETF directions. We address this issue by introducing a proximal term to the problem's objective function. This guarantees the uniqueness of the solution and stabilises the training process, ensuring that our problem converges to a solution closer to the previous one.

So, the original problem in Equation 6 is transformed into:

$$\underset{\boldsymbol{U} \in St_C^d}{\text{minimize}} \left\| \tilde{\boldsymbol{H}} - \boldsymbol{U}\tilde{\boldsymbol{M}} \right\|_F^2 + \frac{\delta}{2} \left\| \boldsymbol{U} - \boldsymbol{U}_{\text{prox}} \right\|_F^2 . \tag{7}$$

Here, $\boldsymbol{U}_{\text{prox}}$ represents the proximal target simplex ETF direction, and $\delta > 0$ serves as the proximal coefficient, handling the trade-off between achieving the optimal solution's proximity to the feature means and its proximity to a given simplex ETF direction. In fact, one can perceive our problem formulation in Equation 7 as a generalisation to a predetermined fixed simplex ETF solution. This is evident when considering that if we significantly increase $\delta$, the optimal direction $\boldsymbol{U}^\star$ would converge towards the fixed proximal direction $\boldsymbol{U}_{\text{prox}}$.

## 3.4 General Learning Setting

Our problem formulation, following the general deep neural network architecture in Equation 3, can be seen as a bilevel optimisation problem as follows:

$$\underset{\boldsymbol{\Theta}}{\text{minimize}} \ \mathcal{L}(\mathcal{D}; \boldsymbol{\Theta}, \boldsymbol{U}^\star) \triangleq -\frac{1}{N} \sum_{i=1}^{N} \log \left( \frac{\exp\left(\psi_{\boldsymbol{\Theta}}(\boldsymbol{x}_i, \boldsymbol{U}^\star)_{y_i}\right)}{\sum_{j=1}^{C} \exp\left(\psi_{\boldsymbol{\Theta}}(\boldsymbol{x}_i, \boldsymbol{U}^\star)_j\right)} \right) ,$$

$$\text{subject to} \ \boldsymbol{U}^\star \in \underset{\boldsymbol{U} \in St_C^d}{\arg\min} \left\| \tilde{\boldsymbol{H}} - \boldsymbol{U}\tilde{\boldsymbol{M}} \right\|_F^2 + \frac{\delta}{2} \left\| \boldsymbol{U} - \boldsymbol{U}_{\text{prox}} \right\|_F^2 , \tag{8}$$

where $\psi_{\boldsymbol{\Theta}}(\boldsymbol{x}_i, \boldsymbol{U}^\star) = \tau \boldsymbol{M}\boldsymbol{U}^\star(\boldsymbol{h}_i - \boldsymbol{h}_G)$ with $\boldsymbol{h}_i = \phi_{\boldsymbol{\theta}}(\boldsymbol{x}_i)$. Here, $\psi$ denotes the logits, where the classifier weights are set as $\boldsymbol{W} = \boldsymbol{M}\boldsymbol{U}^\star$, and the bias is set to $\boldsymbol{b} = -\boldsymbol{M}\boldsymbol{U}^\star\boldsymbol{h}_G$ to account for feature centring. Furthermore, $\boldsymbol{M}$ is the standard simplex ETF, $\tilde{\boldsymbol{M}}$ is its normalised version, and $\tilde{\boldsymbol{H}}$ is the normalised centred feature matrix. The temperature parameter $\tau > 0$ controls the lower bound of the cross-entropy loss when dealing with normalised features, as defined in [69, Theorem 1].

## 3.5 Handling Stochastic Updates

In practice, we use stochastic gradient descent updates, and, as such, adjustments to our computations are necessary. With each gradient update now based on a mini-batch, we implement two key

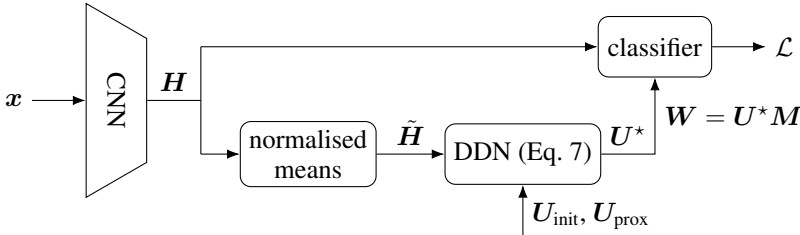

Figure 1: Schematic of our proposed architecture for optimising towards the nearest simplex ETF. The classifier weights $\boldsymbol{W} = \boldsymbol{U}^{\star}\boldsymbol{M}$ are an implicit function of the CNN features $\boldsymbol{H}$. Note that the parameters of the CNN are updated via two gradient paths from the loss function $\mathcal{L}$, a direct path (top) and an indirect path through $\boldsymbol{U}^{\star}$ (bottom).

changes. First, rather than directly optimising the problem of finding the nearest simplex ETF geometry concerning the feature means of the mini-batch, we introduce an exponential moving average operation during the computation of the feature means. This operation accumulates statistics and enhances training stability throughout iterations. Formally, at time step $t$, we have the following equation, where $\alpha \in \mathbb{R}$ represents the smoothing factor:

$$\tilde{\boldsymbol{H}}_t = \alpha\tilde{\boldsymbol{H}}_{\text{batch}} + (1 - \alpha)\tilde{\boldsymbol{H}}_{t-1} \ . \tag{9}$$

Second, we employ stratified batch sampling to guarantee that all class labels are represented in the mini-batch. This ensures that we avoid degenerate solutions when finding the nearest simplex ETF geometry, as our optimisation problem requires input feature means for all $C$ classes. In cases where the number of classes exceeds the chosen batch size, we compute the per-class feature mean for the class labels present in the given batch. For the remaining class labels, we set their feature mean as the global mean of the batch. We repeat this process for each training iteration until we have sampled examples belonging to the missing class labels. At that point, we update the feature mean of those missing class labels with the new feature statistics. We reserve this method only for cases where the batch size is smaller than the number of labels since it can introduce instability during early iterations.

## 3.6 Deep Declarative Layer

We can backpropagate through the Riemannian optimisation problem to update the feature means using a declarative node [23]. Then, the features are updated from both the loss and the feature means through auto-differentiation. The motivation for developing the DDN layer lies in recognising that, despite the presence of a proximal term, abrupt and sudden changes to the classifier may occur as the features are updated. These changes can pose challenges for backpropagation, potentially disrupting the stability and convergence of the training process. Incorporating an additional stream of gradients through the feature means to account for such changes, as depicted in Figure 1, assists in stabilising the feature updates during backpropagation.

To efficiently backpropagate through the optimisation problem, we employ techniques described in Gould et al. [23] utilising the implicit function theorem to compute the gradients. In our case, we have a scalar objective function $f : \mathbb{R}^{d \times C} \to \mathbb{R}$, and a matrix constraint function $J : \mathbb{R}^{d \times C} \to \mathbb{R}^{C \times C}$. Since we have matrix variables, we use vectorisation techniques [46] to avoid numerically dealing with tensor gradients. More specifically, we have the following:

**Proposition 1** (Following directly from Proposition 4.5 in Gould et al. [23])**.** *Consider the optimisation problem in Equation 7. Assume that the solution exists and that the objective function $f$ and the constraint function $J$ are twice differentiable in the neighbourhood of the solution. If the* $\text{rank}(\boldsymbol{A}) = \frac{C(C+1)}{2}$ *and $\boldsymbol{G}$ is non-singular then:*

$$Dy(\boldsymbol{U}) = \boldsymbol{G}^{-1}\boldsymbol{A}^{\top}(\boldsymbol{A}\boldsymbol{G}^{-1}\boldsymbol{A}^{\top})^{-1}(\boldsymbol{A}\boldsymbol{G}^{-1}\boldsymbol{B}) - \boldsymbol{G}^{-1}\boldsymbol{B} \ ,$$

*where,*

$$\boldsymbol{A} = \text{rvech}(D_{\boldsymbol{U}}J(\tilde{\boldsymbol{H}}, \boldsymbol{U})) \in \mathbb{R}^{\frac{C(C+1)}{2} \times dC} \ ,$$

$$\boldsymbol{B} = \text{rvec}(D^2_{\tilde{\boldsymbol{H}}\boldsymbol{U}} f(\tilde{\boldsymbol{H}}, \boldsymbol{U})) \in \mathbb{R}^{dC \times dC} \ ,$$

$$\boldsymbol{G} = \text{rvec}(D^2_{\boldsymbol{U}\boldsymbol{U}} f(\tilde{\boldsymbol{H}}, \boldsymbol{U})) - \boldsymbol{\Lambda} : \text{rvech}(D^2_{\boldsymbol{U}\boldsymbol{U}} J(\tilde{\boldsymbol{H}}, \boldsymbol{U})) \in \mathbb{R}^{dC \times dC} \ .$$

*Here, the double dot product symbol* $(:)$ *denotes a tensor contraction on appropriate indices between the Lagrange multiplier matrix* $\boldsymbol{\Lambda}$ *and a fourth-order tensor Hessian. Also,* $\mathrm{rvec}(\cdot)$ *and* $\mathrm{rvech}(\cdot)$ *refer to the row-major vectorisation and half-vectorisation operations, respectively. To find the Lagrange multiplier matrix* $\boldsymbol{\Lambda} \in \mathbb{R}^{C \times \frac{C+1}{2}}$, *we solve the following equation where we have vectorised the matrix as* $\boldsymbol{\lambda} \in \mathbb{R}^{\frac{C(C+1)}{2}}$,

$$\boldsymbol{\lambda}^{\top} \boldsymbol{A} = D_{\boldsymbol{U}}\, f(\tilde{\boldsymbol{H}}, \boldsymbol{U})\,.$$

*Alternatively, for a more efficient computation of the identity* $\boldsymbol{G}$, *we can utilise the embedded gradient field method as defined in Birtea et al. [9, 10]. Therefore, we obtain:*

$$\boldsymbol{G} = \mathrm{rvec}(D_{\boldsymbol{UU}}^2\, f(\tilde{\boldsymbol{H}}, \boldsymbol{U})) - \boldsymbol{I}_d \otimes \Sigma(\boldsymbol{U}) \in \mathbb{R}^{dC \times dC}\,,$$

*where* $\Sigma(\boldsymbol{U}) = \frac{1}{2}\Big(D_{\boldsymbol{U}}\, f(\tilde{\boldsymbol{H}}, \boldsymbol{U})^{\top} \boldsymbol{U} + \boldsymbol{U}^{\top} D_{\boldsymbol{U}}\, f(\tilde{\boldsymbol{H}}, \boldsymbol{U})\Big)$, *and* $\otimes$ *here denotes Kronecker product.*

A detailed derivation of each identity in the proposition can be found in Appendix B.

## 4  Experiments

In our experiments, we perform feature normalisation onto a hypersphere, a common practice in training neural networks, which improves representation and enhances model performance [71, 26, 53, 42, 61, 62, 12, 16, 35]. We find that combining classifier weight normalisation with feature normalisation accelerates convergence [69]. Given that simplex ETFs are inherently normalised, we include classifier weight normalisation in our standard training procedure to ensure fair method comparisons.

**Experimental Setup.**   In this study, we conduct experiments on three model variants. First, the standard method involves training a model with learnable classifier weights, following conventional practice. Second, in the fixed ETF method, we set the classifier to a predefined simplex ETF. In all experiments, we choose the simplex ETF with canonical direction. In Appendix C, we also include additional experiments for fixed simplex ETFs with random directions generated from a Haar measure [47]. Last, our implicit ETF method, where we set the classifier weights on-the-fly as the simplex ETF closest to the current feature means.

We repeat experiments on each method five times with distinct random seeds and report the median values alongside their respective ranges. For reproducibility and to streamline hyperparameter tuning, we employed Automatic Gradient Descent (AGD) [6]. Following the authors' recommendation, we set the gain/momentum parameter to 10 to expedite convergence, aligning it with other widely used optimisers like Adam [36] and SGD. Our experiments on real datasets run for 200 epochs with batch size 256; for the UFM analysis, we run 2000 iterations.

Our method underwent rigorous evaluation across various UFM sizes and real model architectures trained on actual datasets, including CIFAR10 [37], CIFAR100 [37], STL10 [14], and ImageNet-1000 [15], implemented on ResNet [29] and VGG [56] architectures. More specifically, we trained CIFAR10 on ResNet18 and VGG13, CIFAR100 and STL10 on ResNet50 and VGG13, and ImageNet-1000 on ResNet50. The input images were preprocessed pixel-wise by subtracting the mean and dividing by the standard deviation. Additionally, standard data augmentation techniques were applied, including random horizontal flips, rotations, and crops. All experiments were conducted using Nvidia RTX3090 and A100 GPUs.

**Hyperparameter Selection and Riemannian Initialisation Schemes.**   We solve the Riemannian optimisation problem defined in Equation 7 using a Riemannian Trust-Region method [1] from pyManopt [60]. We maintain a proximal coefficient $\delta$ set to $10^{-3}$ consistently across all experiments. It is worth mentioning that algorithm convergence is robust to the precise value of $\delta$. In our problem, determining values for $\boldsymbol{U}_{\mathrm{init}}$ and $\boldsymbol{U}_{\mathrm{prox}}$ is crucial. We explored several methods to initialise these parameters. One approach involved setting both towards the canonical simplex ETF direction. This means initialising them as a partial orthogonal matrix where the first $C$ rows and columns form an identity matrix while the remaining $d - C$ rows are filled with zeros. Another approach is to initialise both of them as random orthogonal matrices from classical compact groups, selected according to a Haar measure [47]. In the end, the approach that yielded the most stable results at initialisation was

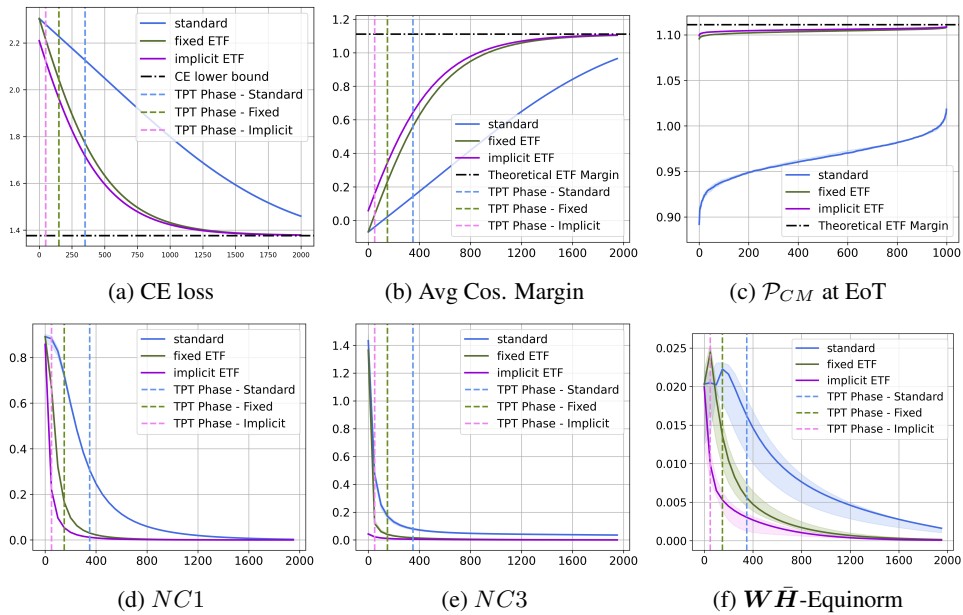

Figure 2: UFM-10 results. In all plots, the x-axis represents the number of epochs, except for plot (c), where the x-axis denotes the number of training examples.

to employ either of the aforementioned initialisation methods to solve the original problem without the proximal term in Equation 6. We then used the obtained $U^\star$ to initialise both $U_{\text{init}}$ and $U_{\text{prox}}$ for the problem in Equation 7. This process was carried out only for the first gradient update of the first epoch. In subsequent iterations, we update these parameters to the $U^\star$ obtained from the previous time step. Importantly, the proximal term is held fixed during each Riemannian optimisation.

Regarding the calculation of the exponential moving average of the feature means, we have found that employing a decay policy on the smoothing factor $\alpha$ yields optimal results. Specifically, we set $\alpha = 2/(T+1)$, where $T$ represents the number of iterations. Additionally, we include a thresholding value of $10^{-4}$, such that if $\alpha$ falls below this threshold, we fix $\alpha$ to be equal to the threshold. This precaution ensures that $\alpha$ does not diminish throughout the iterations, thereby guaranteeing that the newly calculated feature means contribute sufficient statistics to the exponential moving average.

Finally, in our experiments, we set the temperature parameter $\tau$ to five. This choice aligns with the findings discussed by Yaras et al. [69], highlighting the influence of the temperature parameter value on the extent of neural collapse statistics with normalised features.

**Unconstrained Feature Models (UFMs).** Our experiments on UFMs, which provide a controlled setting for evaluating the effectiveness of our method, are done using the following configurations:

- UFM-10: a 10-class UFM containing 1000 features with a dimension of 512.
- UFM-100: a 100-class UFM containing 5000 features with a dimension of 1024.
- UFM-200: a 200-class UFM containing 5000 features, with a dimension of 1024.
- UFM-1000: a 1000-class UFM containing 10000 features, with a dimension of 1024.

**Results.** We present the results for the synthetic UFM-10 case in Figure 2. The CE loss plot demonstrates that fixing the classifier weights to a simplex ETF achieves the theoretical lower bound of Yaras et al. [69, Thm. 1], indicating the attainment of a globally optimal solution. We also visualise the average cosine margin per epoch and the cosine margin distributions of each example at the end of training, defined in Zhou et al. [74]. The neural collapse metrics, $NC1$ and $NC3$, which measure the features' within-class variability, and the self-duality alignment between the feature means and the classifier weights [76], are also plotted. Last, we depict the absolute difference of the classifier and feature means norms to illustrate their convergence towards equinorms, as described in Papyan et al. [50]. A comprehensive description of the metrics can be found in Appendix A. Collectively, the plots indicate the superior performance of our method in achieving a neural collapse (NC) solution

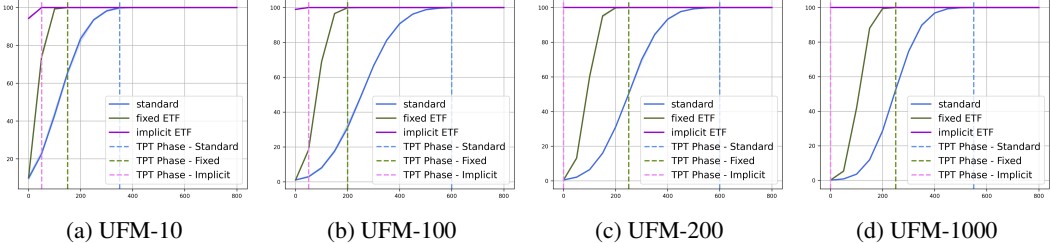

| (a) UFM-10 | (b) UFM-100 | (c) UFM-200 | (d) UFM-1000 |

Figure 3: The evolution of convergence measured in top-1 accuracy of the UFM as we increase the number of classes, plotted for the first 800 epochs. We omit the rest of the epochs as all methods have converged and have identical results.

Table 1: Train top-1 accuracy results presented as a median with indices indicating the range of values from five random seeds. Best results are bolded.

| Dataset | Network | Train accuracy at epoch 50 | | | Train accuracy at epoch 200 | | |
|---|---|---|---|---|---|---|---|
| | | Standard | Fixed ETF | Implicit ETF | Standard | Fixed ETF | Implicit ETF |
| CIFAR10 | ResNet18 | $87.42_{86.1}^{89.7}$ | $86.89_{84.7}^{88.6}$ | $\mathbf{88.71}_{88.5}^{89.6}$ | $96.69_{96.5}^{98.6}$ | $97.18_{95.6}^{97.9}$ | $\mathbf{98.09}_{97.9}^{98.6}$ |
| | VGG13 | $93.59_{90.7}^{97.0}$ | $76.66_{53.9}^{85.8}$ | $\mathbf{95.69}_{95.2}^{96.2}$ | $99.15_{97.9}^{99.8}$ | $79.36_{59.6}^{99.1}$ | $\mathbf{99.56}_{99.4}^{99.7}$ |
| CIFAR100 | ResNet50 | $58.47_{53.9}^{59.6}$ | $63.93_{61.1}^{65.2}$ | $\mathbf{72.15}_{70.1}^{74.1}$ | $95.87_{94.7}^{98.6}$ | $91.34_{90.4}^{92.1}$ | $\mathbf{96.96}_{96.2}^{97.3}$ |
| | VGG13 | $82.00_{80.5}^{84.0}$ | $81.14_{76.0}^{81.9}$ | $\mathbf{88.39}_{86.9}^{89.4}$ | $99.34_{99.2}^{99.6}$ | $94.55_{92.5}^{95.3}$ | $98.92_{98.8}^{99.0}$ |
| STL10 | ResNet50 | $83.86_{84.5}^{90.7}$ | $86.76_{77.8}^{86.8}$ | $\mathbf{93.54}_{91.3}^{95.3}$ | $99.42_{99.0}^{99.9}$ | $98.38_{98.1}^{99.3}$ | $\mathbf{99.72}_{98.0}^{99.9}$ |
| | VGG13 | $82.66_{73.7}^{90.7}$ | $83.60_{65.6}^{85.1}$ | $\mathbf{90.14}_{69.2}^{93.5}$ | $100.0_{100}^{100}$ | $99.92_{99.8}^{99.9}$ | $\mathbf{100.0}_{100}^{100}$ |
| ImageNet | ResNet50 | $58.35_{58.0}^{59.1}$ | $70.44_{69.5}^{70.7}$ | $\mathbf{74.09}_{73.8}^{74.5}$ | $77.20_{76.5}^{77.3}$ | $83.09_{83.1}^{83.6}$ | $\mathbf{88.01}_{87.5}^{88.5}$ |

faster than other approaches. In Figure 3, we demonstrate under the UFM setting that as we increase the number of classes, our method maintains constant performance and converges at the same rate, while the fixed ETF and the standard approach require more time to reach the interpolation threshold.

Numerical results for the top-1 train and test accuracy are reported in Tables 1 and 2, respectively. The results are provided for snapshots taken at epoch 50 and epoch 200. It is evident that our method achieves a faster convergence speed compared to the competitive methods while ultimately converging to the same performance level. Additionally, it is noteworthy that our method exhibits the smallest degree of variability across different runs, as indicated by the range values provided. Finally, in Figure 4, we present qualitative results that confirm our solution's ability to converge much faster and reach peak performance earlier than the standard and fixed ETF methods on ImageNet. It's important to note that the standard method with AGD is reported to converge to the same testing accuracy ($65.5\%$) at epoch 350, as shown in Bernstein et al. [6, Figure 4]. At epoch 200, the authors exhibit a testing accuracy of approximately $51\%$. Since we have increased the gain parameter on AGD compared to the results reported in the original paper, we report a final $60.67\%$ testing accuracy for the standard method, whereas our method reaches peak convergence at approximately epoch 80. We note that the ImageNet results reported in Tables 1 and 2, as well as Figure 4, are generated solely by solving the Riemannian optimisation problem without considering its gradient stream on the feature updates, due to computational constraints. We discuss the computational requirements of our method in Section 5. We also present qualitative results for all the other datasets and architectures in Appendix C.

## 5 Discussion: Limitations and Future Directions

Our method involves two gradient streams updating the features, as depicted in Figure 1. Interestingly, empirical observations on small-scale datasets (see Figure 15) indicate that even without the back-propagation through the DDN layer, the performance remains comparable, rendering the gradient calculation of the DDN layer optional. In Figure 15c, we observe a strong impact of the DDN layer gradient on the atomic feature level, with more features reaching the theoretical simplex ETF margin by the end of training. To reach a consensus on the exact effect of the DDN gradient on the learning

Table 2: Test top-1 accuracy results presented as a median with indices indicating the range of values from five random seeds. Best results are bolded.

| Dataset | Network | Test accuracy at epoch 50 | | | Test accuracy at epoch 200 | | |
|---|---|---|---|---|---|---|---|
| | | Standard | Fixed ETF | Implicit ETF | Standard | Fixed ETF | Implicit ETF |
| CIFAR10 | ResNet18 | $80.47^{82.6}_{79.4}$ | $80.63^{81.8}_{79.4}$ | $\mathbf{81.76}^{82.0}_{81.4}$ | $83.97^{84.8}_{83.2}$ | $84.53^{84.9}_{83.7}$ | $\mathbf{84.78}^{85.1}_{84.3}$ |
| | VGG13 | $86.70^{89.4}_{83.7}$ | $70.99^{80.7}_{50.7}$ | $\mathbf{88.30}^{88.7}_{87.4}$ | $90.34^{91.5}_{89.1}$ | $72.48^{90.5}_{56.9}$ | $\mathbf{90.98}^{91.5}_{90.6}$ |
| CIFAR100 | ResNet50 | $45.91^{46.3}_{42.1}$ | $45.37^{45.6}_{43.2}$ | $\mathbf{48.38}^{49.3}_{48.0}$ | $\mathbf{51.29}^{51.9}_{50.4}$ | $48.03^{49.3}_{47.7}$ | $50.52^{51.2}_{50.2}$ |
| | VGG13 | $60.82^{61.2}_{59.3}$ | $60.10^{61.1}_{57.1}$ | $\mathbf{62.39}^{63.6}_{61.8}$ | $\mathbf{67.54}^{68.1}_{66.4}$ | $62.78^{63.4}_{61.6}$ | $67.14^{67.6}_{66.8}$ |
| STL10 | ResNet50 | $55.41^{58.3}_{54.8}$ | $\mathbf{58.11}^{60.0}_{57.1}$ | $57.56^{59.2}_{56.4}$ | $63.60^{65.2}_{61.8}$ | $\mathbf{64.33}^{66.5}_{63.9}$ | $62.75^{63.0}_{60.7}$ |
| | VGG13 | $66.09^{72.3}_{60.0}$ | $66.15^{67.7}_{52.2}$ | $\mathbf{68.78}^{71.3}_{56.2}$ | $\mathbf{81.61}^{81.9}_{81.3}$ | $79.53^{80.3}_{78.6}$ | $79.94^{80.9}_{79.5}$ |
| ImageNet | ResNet50 | $52.64^{53.3}_{52.3}$ | $58.85^{59.1}_{58.3}$ | $\mathbf{63.40}^{63.8}_{63.2}$ | $60.02^{60.7}_{59.8}$ | $61.47^{62.0}_{61.4}$ | $\mathbf{65.36}^{65.7}_{65.0}$ |

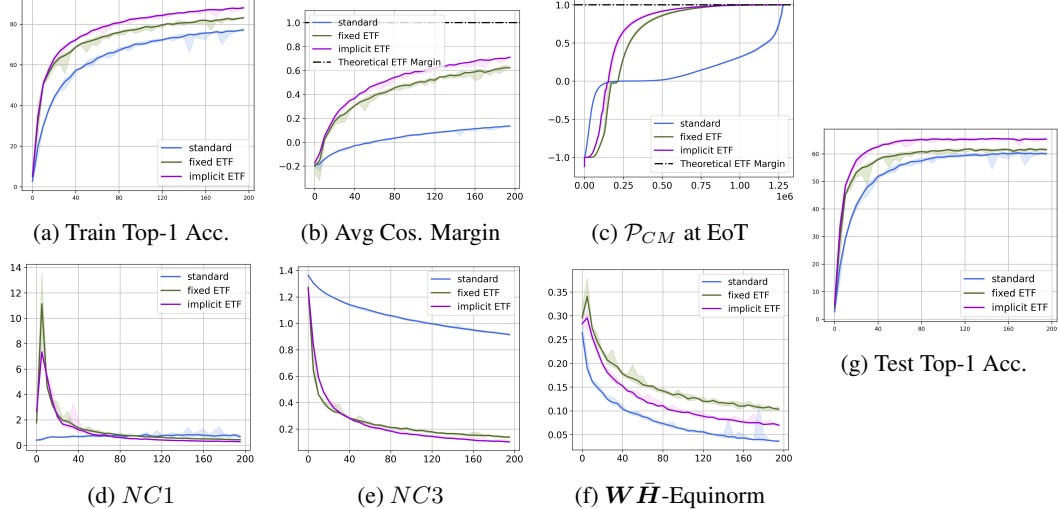

(a) Train Top-1 Acc.  (b) Avg Cos. Margin  (c) $\mathcal{P}_{CM}$ at EoT

(d) $NC1$  (e) $NC3$  (f) $\boldsymbol{W}\bar{\boldsymbol{H}}$-Equinorm  (g) Test Top-1 Acc.

Figure 4: ImageNet results on ResNet-50. In all plots, the x-axis represents the number of epochs, except for plot (c), where the x-axis denotes the number of training examples.

process, further experiments on large-scale datasets are needed. However, on large-scale datasets with large $d$ and $C$, such as ImageNet, computing the backward pass of the Riemannian optimisation is challenging due to the memory inefficiency of the current implementation of DDN gradients. This limitation is an area we aim to address in future work. Note that in all other experiments, we use the full gradient computations, including both direct and indirect components, through the DDN layer. We summarise the GPU memory requirements for each method across various datasets in Table 3.

Our discussion so far has focused on convergence speed in terms of the number of epochs required for the network to converge. However, it is also important to consider the time required per epoch. In our case, as training progresses, the time taken by the Riemannian optimization quickly becomes almost negligible compared to the network's total forward pass time, while it approaches the standard and fixed ETF training forward times, as shown in Figure 5a. However, DDN gradient computation increases considerably when the feature dimension $d$ and the number of classes $C$ increase and starts to dominate the runtime for large datasets such as ImageNet. Nevertheless, for ImageNet, we do not compute the DDN gradients and still outperform other methods. We plan to explore ways to expedite the DDN forward and backward pass in future work.

## 6 Conclusion

In this paper, we introduced a novel method for determining the nearest simplex ETF to the penultimate features of a neural network and utilising it as our target classifier at each iteration. This contrasts with previous approaches, which either fix to a specific simplex ETF or allow the network

Table 3: GPU memory (in Gigabytes) during training.

| Model | Standard | Fixed ETF | Implicit ETF w/o DDN Bwd | w/ DDN Bwd |
|---|---|---|---|---|
| UFM-10 | 1.5 | 1.5 | 1.5 | 1.6 |
| UFM-100 | 1.7 | 1.7 | 1.7 | 10.7 |
| UFM-200 | 1.7 | 1.7 | 1.8 | 70.3 |
| UFM-1000 | 1.9 | 1.9 | 2.9 | N/A |
| ResNet18 - CIFAR10 | 2.2 | 2.2 | 2.2 | 2.3 |
| ResNet50 - CIFAR10 | 2.6 | 2.6 | 2.7 | 2.8 |
| ResNet50 - CIFAR100 | 2.6 | 2.6 | 2.7 | 18.9 |
| ResNet50 - ImageNet | 27.5 | 27.2 | 27.8 | N/A |

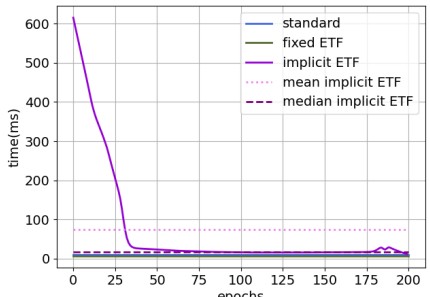

(a) Forward pass times in milliseconds.

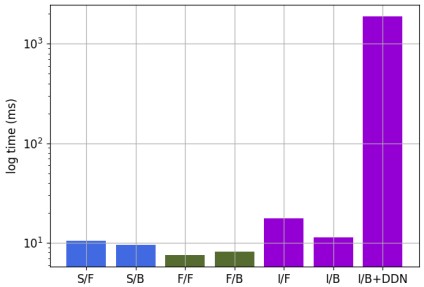

(b) Forward and backward times in (log) millisecs.

Figure 5: CIFAR100 computational cost results on ResNet-50. In (a), we plot the forward pass time for each method. For the implicit ETF method, which has dynamic computation times, we also include the mean and median time values. In (b), we plot the computational cost for each forward and backward pass across methods. For the implicit ETF forward pass, we have taken its median time. The notation is as follows: S/F = Standard Forward Pass, S/B = Standard Backward Pass, F/F = Fixed ETF Forward Pass, F/B = Fixed ETF Backward Pass, I/F = Implicit ETF Forward Pass, and I/B = Implicit ETF Backward Pass.

to learn it through gradient descent. Our method involves solving a Riemannian optimisation problem facilitated by a deep declarative node, enabling backpropagation through this process.

We demonstrated that our approach enhances convergence speed across various datasets and architectures while also reducing variability stemming from different random initialisations. By defining the optimal structure of the classifier and efficiently leveraging its rotation invariance property to find the one closest to the backbone features, we anticipate that our method will facilitate the creation of new architectures and the utilisation of new datasets without necessitating specific learning or tuning of the classifier's structure.

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

## Appendices

## A   Background

Following the definitions of the global mean and class mean of the penultimate-layer features $\{\boldsymbol{h}_{c,i}\}$ in Equation 4, here we introduce the within-class and between-class covariances,

$$\boldsymbol{\Sigma_W} \triangleq \frac{1}{N} \sum_{c=1}^{C} \sum_{i=1}^{n_c} \left(\boldsymbol{h}_{c,i} - \bar{\boldsymbol{h}}_c\right) \left(\boldsymbol{h}_{c,i} - \bar{\boldsymbol{h}}_c\right)^{\top} ,$$

$$\boldsymbol{\Sigma_B} \triangleq \frac{1}{C} \sum_{c=1}^{C} \left(\bar{\boldsymbol{h}}_c - \boldsymbol{h}_G\right) \left(\bar{\boldsymbol{h}}_c - \boldsymbol{h}_G\right)^{\top} . \tag{10}$$

We proceed by expanding on the four key properties of the last-layer activations and classifiers, as empirically observed by Papyan et al. [50] at the terminal phase of training (TPT), where we have achieved zero classification error and continue towards zero loss.

**NC1**  Variability Collapse: Throughout training, feature activation variability diminishes as they converge towards their respective class means.

$$NC1 \triangleq \frac{1}{C} \operatorname{Tr}\left(\boldsymbol{\Sigma_W} \boldsymbol{\Sigma_B^{\dagger}}\right) , \tag{11}$$

where † denotes the Moore–Penrose inverse.

**NC2**  Convergence to Simplex ETF: The class-mean activation vectors, centred around their global mean, converge to uniform norms while simultaneously maintaining equal-sized and maximally separable angles between them[2].

$$NC2 \triangleq \left\| \frac{\boldsymbol{W}\boldsymbol{W}^{\top}}{\|\boldsymbol{W}\boldsymbol{W}^{\top}\|_F} - \frac{1}{\sqrt{C-1}} \left(\boldsymbol{I}_C - \frac{1}{C}\boldsymbol{1}_C\boldsymbol{1}_C^{\top}\right) \right\|_F . \tag{12}$$

**NC3**  Convergence to Self-duality: The feature class-means and linear classifiers eventually align in a dual vector space up to some scaling.

$$NC3 \triangleq \left\| \frac{\boldsymbol{W}\bar{\boldsymbol{H}}}{\|\boldsymbol{W}\bar{\boldsymbol{H}}\|_F} - \frac{1}{\sqrt{C-1}} \left(\boldsymbol{I}_C - \frac{1}{C}\boldsymbol{1}_C\boldsymbol{1}_C^{\top}\right) \right\|_F . \tag{13}$$

**NC4**  Simplification to Nearest Class-Center (NCC): The network classifier tends to select the class whose mean is closest (in Euclidean distance) to a given deepnet activation.

$$NC4 \triangleq \arg\max_{c'}\langle\boldsymbol{w}_{c'}, \boldsymbol{h}\rangle + b_{c'} \to \arg\min_{c'} \|\boldsymbol{h} - \bar{\boldsymbol{h}}_{c'}\|_2 . \tag{14}$$

Since the NC2 and NC3 metrics involve normalised matrices, it is not immediately evident whether the linear classifier and the class-mean activations are equinorm. Consequently, we introduce the following definition:

$$\boldsymbol{W}\bar{\boldsymbol{H}}\text{-Equinorm} = |W_{\text{equinorm}} - \bar{H}_{\text{equinorm}}| , \tag{15}$$

where $\bar{H}_{\text{equinorm}} = \frac{\text{std}_c(\|\bar{\boldsymbol{h}}_c - \boldsymbol{h}_G\|_2)}{\text{avg}_c(\|\bar{\boldsymbol{h}}_c - \boldsymbol{h}_G\|_2)}$ and $W_{\text{equinorm}} = \frac{\text{std}_c(\|\boldsymbol{w}_c\|_2)}{\text{avg}_c(\|\boldsymbol{w}_c\|_2)}$.

Finally, to measure the extent of the variability collapse of each feature separately, we define the cosine margin metric as follows:

$$CM_{c,i} = \cos\theta_{c,i;j} - \max_{j\neq c}\cos\theta_{c,i;j}, \quad \text{where } \cos\theta_{c,i;j} = \frac{\langle\boldsymbol{w}_j - \boldsymbol{w}_G, \boldsymbol{h}_{c,i} - \boldsymbol{h}_G\rangle}{\|\boldsymbol{w}_j - \boldsymbol{w}_G\|_2\|\boldsymbol{h}_{c,i} - \boldsymbol{h}_G\|_2} . \tag{16}$$

Note that the maximal angle for a simplex ETF vector collection $\{v_c\}_{c=1}^{C}$ are defined as such:

$$\frac{\langle v_c, v_{c'}\rangle}{\|v_c\|_2\|v_{c'}\|_2} = \begin{cases} 1, & \text{for } c = c' \\ -\frac{1}{C-1}, & \text{for } c \neq c' \end{cases} . \tag{17}$$

Therefore, we can calculate the theoretical simplex ETF cosine margin as $\frac{C}{C-1}$.

---

[2]Mathematically, we have defined NC2 as the collapse of the linear classifiers to a simplex ETF.

# B DDN Gradients

The original problem we are trying to solve, as defined in Section 3.2, is the following:

$$\underset{\boldsymbol{U} \in St_C^d}{\text{minimize}} \left\| \tilde{\boldsymbol{H}} - \boldsymbol{U}\tilde{\boldsymbol{M}} \right\|_F^2 , \tag{18}$$

or equivalently expanded as such:

$$y(\boldsymbol{U}) \in \underset{\boldsymbol{U} \in \mathbb{R}^{d \times C}}{\arg\min} \quad \tilde{\boldsymbol{H}} : \tilde{\boldsymbol{H}} - \frac{2}{\sqrt{C-1}} \tilde{\boldsymbol{H}} : \boldsymbol{U}\tilde{\boldsymbol{M}} + \frac{1}{C-1} \boldsymbol{U} : \boldsymbol{U}\tilde{\boldsymbol{M}} ,$$
$$\text{subject to} \quad \boldsymbol{U}^\top \boldsymbol{U} = \boldsymbol{I}_C . \tag{19}$$

Here, we denote the double-dot operator : as the Frobenius inner product, *i.e.*, $\boldsymbol{A} : \boldsymbol{B} = \sum_{i=1}^m \sum_{j=1}^n A_{ij} B_{ij} = \text{Tr}(\boldsymbol{A}^\top \boldsymbol{B})$.

To compute the first and second-order gradients of the objective function and constraints, respectively, we need to consider matrices as variables. Utilising matrix differentials and vectorised derivatives, as defined in [46], simplifies the computation of second-order objective derivatives and all constraint derivatives. For the objective function, we have:

$$d f(\tilde{\boldsymbol{H}}, \boldsymbol{U}) = -\frac{2}{\sqrt{C-1}} \tilde{\boldsymbol{H}} : d\boldsymbol{U}\tilde{\boldsymbol{M}} + \frac{1}{C-1} \left( d\boldsymbol{U} : \boldsymbol{U}\tilde{\boldsymbol{M}} + \boldsymbol{U} : d\boldsymbol{U}\tilde{\boldsymbol{M}} \right)$$

$$= \frac{2}{\sqrt{C-1}} \tilde{\boldsymbol{H}}\tilde{\boldsymbol{M}} : d\boldsymbol{U} + \frac{2}{C-1} \boldsymbol{U}\tilde{\boldsymbol{M}} : d\boldsymbol{U}$$

$$= \left( \frac{2}{\sqrt{C-1}} \tilde{\boldsymbol{H}}\tilde{\boldsymbol{M}} + \frac{2}{C-1} \boldsymbol{U}\tilde{\boldsymbol{M}} \right) : d\boldsymbol{U} \tag{20}$$

$$\implies D_{\boldsymbol{U}} f(\tilde{\boldsymbol{H}}, \boldsymbol{U}) = \frac{2}{\sqrt{C-1}} \tilde{\boldsymbol{H}}\tilde{\boldsymbol{M}} + \frac{2}{C-1} \boldsymbol{U}\tilde{\boldsymbol{M}} \in \mathbb{R}^{d \times C} .$$

$$d\, D_{\boldsymbol{U}}(\tilde{\boldsymbol{H}}, \boldsymbol{U}) = \frac{2}{C-1} d\boldsymbol{U}\tilde{\boldsymbol{M}}$$

$$\implies d\, \text{rvec}\, D_{\boldsymbol{U}}(\tilde{\boldsymbol{H}}, \boldsymbol{U}) = \frac{2}{C-1} \text{rvec}(d\boldsymbol{U}\tilde{\boldsymbol{M}}) = \frac{2}{C-1} \left( \boldsymbol{I}_d \otimes \tilde{\boldsymbol{M}} \right) d\, \text{rvec}\, \boldsymbol{U} \tag{21}$$

$$\implies \text{rvec}(D_{\boldsymbol{U}\boldsymbol{U}}^2 f(\tilde{\boldsymbol{H}}, \boldsymbol{U})) = \frac{2}{C-1} \left( \boldsymbol{I}_d \otimes \tilde{\boldsymbol{M}} \right) \in \mathbb{R}^{dC \times dC} ,$$

where we have defined here the row-major vectorisation method, *i.e.*, $\text{rvec}(\boldsymbol{A}) = \text{vec}(\boldsymbol{A}^\top)$, and we have used the property:

$$\text{rvec}(\boldsymbol{A}\boldsymbol{B}\boldsymbol{C}) = (\boldsymbol{A} \otimes \boldsymbol{C}^\top)\text{rvec}(\boldsymbol{B}) . \tag{22}$$

A similar proof follows for the second-order partial gradient of the objective. We omit the proof here and just declare the result:

$$\boldsymbol{B} \triangleq \text{rvec}(D_{\tilde{\boldsymbol{H}}\boldsymbol{U}}^2 f(\tilde{\boldsymbol{H}}, \boldsymbol{U})) = -\frac{2}{\sqrt{C-1}} \left( \boldsymbol{I}_d \otimes \tilde{\boldsymbol{M}} \right) \in \mathbb{R}^{dC \times dC} . \tag{23}$$

Regarding the gradients of the constraint function, we have:

$$dJ(\tilde{\boldsymbol{H}}, \boldsymbol{U}) = d(\boldsymbol{U}^\top \boldsymbol{U} - \boldsymbol{I}_C) = (d\boldsymbol{U})^\top \boldsymbol{U} + \boldsymbol{U}^\top d\boldsymbol{U}$$

$$\implies d\,\mathrm{rvec}\, J(\tilde{\boldsymbol{H}}, \boldsymbol{U}) = \mathrm{rvec}((d\boldsymbol{U})^\top \boldsymbol{U}) + \mathrm{rvec}(\boldsymbol{U}^\top d\boldsymbol{U})$$

$$= (\boldsymbol{I}_C \otimes \boldsymbol{U}^\top)\, d\,\mathrm{rvec}\, \boldsymbol{U}^\top + (\boldsymbol{U}^\top \otimes \boldsymbol{I}_C)\, d\,\mathrm{rvec}\, \boldsymbol{U}$$

$$= (\boldsymbol{I}_C \otimes \boldsymbol{U}^\top)\boldsymbol{K}_{dC}\, d\,\mathrm{rvec}\, \boldsymbol{U} + (\boldsymbol{U}^\top \otimes \boldsymbol{I}_C)\, d\,\mathrm{rvec}\, \boldsymbol{U}$$

$$= \boldsymbol{K}_{CC}(\boldsymbol{U}^\top \otimes \boldsymbol{I}_C)\, d\,\mathrm{rvec}\, \boldsymbol{U} + (\boldsymbol{U}^\top \otimes \boldsymbol{I}_C)\, d\,\mathrm{rvec}\, \boldsymbol{U} \tag{24}$$

$$= (\boldsymbol{K}_{CC} + \boldsymbol{I}_{C^2})(\boldsymbol{U}^\top \otimes \boldsymbol{I}_C)\, d\,\mathrm{rvec}\, \boldsymbol{U}$$

$$= (\boldsymbol{K}_{CC} + \boldsymbol{I}_{C^2})(\boldsymbol{U} \otimes \boldsymbol{I}_C)^\top d\,\mathrm{rvec}\, \boldsymbol{U}$$

$$\implies \mathrm{rvec}(D_{\boldsymbol{U}} J(\tilde{\boldsymbol{H}}, \boldsymbol{U})) = (\boldsymbol{K}_{CC} + \boldsymbol{I}_{C^2})(\boldsymbol{U} \otimes \boldsymbol{I}_C)^\top \in \mathbb{R}^{CC \times dC} \,,$$

where we have defined the commutation matrix [46, 13] as $\boldsymbol{K}_{mn}$, as a matrix which satisfies the two following identities for any given $\boldsymbol{A} \in \mathbb{R}^{m \times n}$ and $\boldsymbol{B} \in \mathbb{R}^{r \times q}$:

$$\boldsymbol{K}_{mn}\mathrm{vec}(\boldsymbol{A}) = \mathrm{vec}(\boldsymbol{A}^\top) \,,$$
$$\boldsymbol{K}_{rm}(\boldsymbol{A} \otimes \boldsymbol{B})\boldsymbol{K}_{nq} = \boldsymbol{B} \otimes \boldsymbol{A} \,. \tag{25}$$

The gradient in Equation 24 contains redundant constraints because of the symmetrical nature of the orthogonality constraints. To retain only the non-redundant constraints, we must undertake a half-vectorisation procedure. Given that we already possess the fully vectorised gradients (which are simpler to compute in this scenario), we require an elimination matrix, $\boldsymbol{L}_C \in \mathbb{R}^{\frac{C(C+1)}{2} \times C^2}$, such that

$$\boldsymbol{L}_C\,\mathrm{rvec}(\boldsymbol{A}) = \mathrm{rvech}(\boldsymbol{A}) \,. \tag{26}$$

We can define an elimination matrix explicitly as follows [45]:

$$\boldsymbol{L}_n = \sum_{i \geq j} \boldsymbol{u}_{ij}\mathrm{vec}(\boldsymbol{E}_{ij})^\top = \sum_{i \geq j}(\boldsymbol{u}_{ij} \otimes \boldsymbol{e}_j^\top \otimes \boldsymbol{e}_i^\top) \,, \tag{27}$$

where

$$\boldsymbol{u}_{ij} = \begin{cases} 1 & \text{in position } (j-1)n + i - \frac{1}{2}j(j-1) \\ 0 & \text{elsewhere} \end{cases} \,. \tag{28}$$

Hence,

$$\boldsymbol{A} \triangleq \mathrm{rvech}(D_{\boldsymbol{U}} J(\tilde{\boldsymbol{H}}, \boldsymbol{U})) = \boldsymbol{L}_C(\boldsymbol{K}_{CC} + \boldsymbol{I}_{C^2})(\boldsymbol{U} \otimes \boldsymbol{I}_C)^\top \in \mathbb{R}^{\frac{C(C+1)}{2} \times dC} \,. \tag{29}$$

We have the following useful Corollary for the second-order gradient of the constraint function.

**Corollary 1** (Magnus & Neudecker [46]). *Let $\phi$ be a twice differentiable real-valued function of an $n \times q$ matrix $\boldsymbol{X}$. Then, the following two relationships hold between the second differential and the Hessian matrix of $\phi$ at $\boldsymbol{X}$:*

$$d^2\phi(\boldsymbol{X}) = \mathrm{Tr}(\boldsymbol{A}(d\boldsymbol{X})^\top \boldsymbol{B}d\boldsymbol{X}) \iff H\phi(\boldsymbol{X}) = \frac{1}{2}(\boldsymbol{A}^\top \otimes \boldsymbol{B} + \boldsymbol{A} \otimes \boldsymbol{B}^\top) \,.$$

With the above corollary established, we can have,

$$\begin{aligned} J(\tilde{\boldsymbol{H}}, \boldsymbol{U}) &= \boldsymbol{U}^\top \boldsymbol{U} - \boldsymbol{I}_C \\ dJ(\tilde{\boldsymbol{H}}, \boldsymbol{U}) &= (d\boldsymbol{U})^\top \boldsymbol{U} + \boldsymbol{U}^\top d\boldsymbol{U} \\ d^2 J(\tilde{\boldsymbol{H}}, \boldsymbol{U}) &= (d\boldsymbol{U})^\top d\boldsymbol{U} + (d\boldsymbol{U})^\top d\boldsymbol{U} = 2(d\boldsymbol{U})^\top d\boldsymbol{U} \\ d^2 J(\tilde{\boldsymbol{H}}, \boldsymbol{U})_{ij} &= 2\boldsymbol{e}_i^\top(d\boldsymbol{U})^\top(d\boldsymbol{U})\boldsymbol{e}_j = 2\,\mathrm{Tr}(\boldsymbol{e}_j\boldsymbol{e}_i^\top(d\boldsymbol{U})^\top d\boldsymbol{U}) \,, \end{aligned} \tag{30}$$

and hence, we get:

$$\text{rvec } (D^2_{\boldsymbol{UU}} J(\tilde{\boldsymbol{H}}, \boldsymbol{U})_{ij}) = \boldsymbol{I}_d \otimes (\boldsymbol{e}_i \boldsymbol{e}_j^\top) + \boldsymbol{I}_d \otimes (\boldsymbol{e}_j \boldsymbol{e}_i^\top) \in \mathbb{R}^{dC \times dC} . \tag{31}$$

Then, we repeat the process with the elimination matrix to eliminate the redundant constraints. However, while this is one way to compute the derivative, a more efficient approach exists, namely the *embedded gradient vector field* method, which we outline in the following subsection. For a complete overview of the method, we recommend readers to follow through the works of Birtea et al. [9, 10], Birtea & Comănescu [8, 7].

Finally, the gradients for the proximal problem in Equation 7 are straightforward to compute, with the only change in gradients being the added proximal terms in:

$$D_{\boldsymbol{U}} f(\tilde{\boldsymbol{H}}, \boldsymbol{U}) = \frac{2}{K-1} \boldsymbol{U}\tilde{\boldsymbol{M}} - \frac{2}{\sqrt{K-1}} \tilde{\boldsymbol{H}}\tilde{\boldsymbol{M}} + \delta(\boldsymbol{U} - \boldsymbol{U}_{\text{prox}}) \in \mathbb{R}^{d \times C} ,$$
$$\tag{32}$$
$$\text{rvec}(D^2_{\boldsymbol{UU}} f(\tilde{\boldsymbol{H}}, \boldsymbol{U})) = \frac{2}{K-1}\left(\boldsymbol{I}_d \otimes \tilde{\boldsymbol{M}}\right) + \delta\boldsymbol{I}_{dC} \in \mathbb{R}^{dC \times dC} .$$

### B.1 Implicit Formulation of Lagrange Multipliers on Differentiable Manifolds

An issue arises in Proposition 1 with the expression for $\boldsymbol{G}$,

$$\boldsymbol{G} = \text{rvec}(D^2_{\boldsymbol{UU}} f(\tilde{\boldsymbol{H}}, \boldsymbol{U})) - \boldsymbol{\Lambda} : \text{rvech}(D^2_{\boldsymbol{UU}} J(\tilde{\boldsymbol{H}}, \boldsymbol{U})) \in \mathbb{R}^{dC \times dC} , \tag{33}$$

where both the calculation of the Lagrange multiplier matrix $\boldsymbol{\Lambda}$ (solved via a linear system) and the construction of the fourth-order tensor representing the second-order derivatives of the constraint function are complex and challenging. However, by recognising the manifold structure of the problem, we can reformulate Equation 33 in a simpler and more computationally efficient way. The embedded gradient vector field method offers such a solution [8].

For a general Riemannian manifold $(\mathcal{M}, g)$, we can define the Gram matrix for the smooth functions $f_1, \ldots, f_s, h_1, \ldots, h_r : (\mathcal{M}, g) \to \mathbb{R}$ as follows,

$$\text{Gram}^{(f_1, \ldots, f_s)}_{(h_1, \ldots, h_r)} \triangleq \begin{bmatrix} \langle \nabla h_1, \nabla f_1 \rangle & \ldots & \langle \nabla h_r, \nabla f_1 \rangle \\ \vdots & \ddots & \vdots \\ \langle \nabla h_1, \nabla f_s \rangle & \ldots & \langle \nabla h_r, \nabla f_s \rangle \end{bmatrix} \in \mathbb{R}^{s \times r} . \tag{34}$$

In our problem, we are working with a compact Stiefel manifold, which is an embedded submanifold of $\mathbb{R}^{d \times C}$, and we identify the isomorphism (via vec) between $\mathbb{R}^{d \times C}$ and $\mathbb{R}^{dC}$. A Stiefel manifold $St^d_C = \{\boldsymbol{U} \in \mathbb{R}^{d \times C} \mid \boldsymbol{U}^\top\boldsymbol{U} = \boldsymbol{I}_C\}$ can be characterised by a set of constraint functions, $j_s, j_{pq} : \mathbb{R}^{dC} \to \mathbb{R}$, as follows:

$$j_s(\boldsymbol{u}) = \frac{1}{2}\|\boldsymbol{u}_s\|^2, \qquad 1 \le s \le C ,$$
$$\tag{35}$$
$$j_{pq}(\boldsymbol{u}) = \langle \boldsymbol{u}_p, \boldsymbol{u}_q \rangle, \qquad 1 \le p < q \le C .$$

We consider a smooth cost function $\tilde{f} : St^d_C \to \mathbb{R}$ and define $f : \mathbb{R}^{d \times C} \to \mathbb{R}$ as a smooth extension (or prolongation) of $\tilde{f}$. The embedded gradient vector field (after applying the isomorphism) is defined on the open set $\mathbb{R}^{dC}_{reg} \subset \mathbb{R}^{dC}$ formed with the regular leaves of the constrained function $\boldsymbol{j} : \mathbb{R}^{dC} \to \mathbb{R}^{\frac{C(C+1)}{2}}$ and is tangent to the foliation of this function. The vector field takes the form [9]:

$$\partial f(\boldsymbol{u}) = \nabla f(\boldsymbol{u}) - \sum_{1 \le s \le C} \sigma_s(\boldsymbol{u})\nabla j_s(\boldsymbol{u}) - \sum_{1 \le p < q \le C} \sigma_{pq}(\boldsymbol{u})\nabla j_{pq}(\boldsymbol{u}) , \tag{36}$$

where $\sigma_s, \sigma_{pq}$ are the Lagrange multiplier functions defined as such [7]:

$$\sigma_s(\boldsymbol{u}) = \frac{\det\left(\mathrm{Gram}_{(j_1,\ldots,j_{s-1},f,j_{s+1},\ldots,j_C,j_{12},\ldots,j_{C-1,C})}^{(j_1,\ldots,j_{s-1},j_s,j_{s+1},\ldots,j_C,j_{12},\ldots,j_{C-1,C})}(\boldsymbol{u})\right)}{\det\left(\mathrm{Gram}_{(j_1,\ldots,j_C,j_{12},\ldots,j_{C-1,C})}^{(j_1,\ldots,j_C,j_{12},\ldots,j_{C-1,C})}(\boldsymbol{u})\right)} \triangleq \frac{\det\left(\mathrm{Gram}_s(\boldsymbol{u})\right)}{\det\left(\mathrm{Gram}(\boldsymbol{u})\right)},$$

$$\sigma_{pq}(\boldsymbol{u}) = \frac{\det\left(\mathrm{Gram}_{(j_1,\ldots,j_C,j_{12},\ldots,j_{pq-1},f,j_{pq+1},\ldots,j_{C-1,C})}^{(j_1,\ldots,j_C,j_{12},\ldots,j_{pq-1},j_{pq},j_{pq+1},\ldots,j_{C-1,C})}(\boldsymbol{u})\right)}{\det\left(\mathrm{Gram}_{(j_1,\ldots,j_C,j_{12},\ldots,j_{C-1,C})}^{(j_1,\ldots,j_C,j_{12},\ldots,j_{C-1,C})}(\boldsymbol{u})\right)} \triangleq \frac{\det\left(\mathrm{Gram}_{pq}(\boldsymbol{u})\right)}{\det\left(\mathrm{Gram}(\boldsymbol{u})\right)}.$$

(37)

The Gram matrix in the denominator of the Lagrange multiplier functions can be defined as a block matrix as follows,

$$\mathrm{Gram}(\boldsymbol{u}) = \begin{bmatrix} \boldsymbol{A} & \boldsymbol{C}^\top \\ \boldsymbol{C} & \boldsymbol{B} \end{bmatrix}, \tag{38}$$

where

$$\boldsymbol{A} = \begin{bmatrix} \langle \nabla j_1, \nabla j_1 \rangle & \cdots & \langle \nabla j_C, \nabla j_1 \rangle \\ \vdots & \ddots & \vdots \\ \langle \nabla j_1, \nabla j_C \rangle & \cdots & \langle \nabla j_C, \nabla j_C \rangle \end{bmatrix},$$

$$\boldsymbol{B} = \begin{bmatrix} \langle \nabla j_{12}, \nabla j_{12} \rangle & \cdots & \langle \nabla j_{C-1,C}, \nabla j_{12} \rangle \\ \vdots & \ddots & \vdots \\ \langle \nabla j_{12}, \nabla j_{C-1,C} \rangle & \cdots & \langle \nabla j_{C-1,C}, \nabla j_{C-1,C} \rangle \end{bmatrix},$$

$$\boldsymbol{C} = \begin{bmatrix} \langle \nabla j_1, \nabla j_{12} \rangle & \cdots & \langle \nabla j_C, \nabla j_{12} \rangle \\ \vdots & \ddots & \vdots \\ \langle \nabla j_1, \nabla j_{C-1,C} \rangle & \cdots & \langle \nabla j_C, \nabla j_{C-1,C} \rangle \end{bmatrix}.$$

For $\mathrm{Gram}_s(\boldsymbol{u})$ and $\mathrm{Gram}_{pq}(\boldsymbol{u})$, we can similarly define block decompositions by replacing the appropriate column of the $\mathrm{Gram}(\boldsymbol{u})$ based on the index. For instance, to define $\mathrm{Gram}_s(\boldsymbol{u})$, we replace the column $s$ with $[\langle \nabla f(\boldsymbol{u}), \nabla j_i(\boldsymbol{u}) \rangle]_{i=1}^{C-1,C}$.

It has been proved [9] that if $\boldsymbol{U} \in St_C^d$ is a critical point of the function $\tilde{f}$, *i.e.*, $\partial f(\boldsymbol{u}) = 0$, then $\sigma_s(\boldsymbol{u}), \sigma_{pq}(\boldsymbol{u})$ become the classical Lagrange multipliers.

**Proposition 2** (Lagrange multiplier functions for Stiefel Manifolds). *The Lagrange multiplier functions are described as a function of the constraint functions in Equation 35 in the following way:*

$$\sigma_s(\boldsymbol{u}) = \langle \nabla f(\boldsymbol{u}), \nabla j_s(\boldsymbol{u}) \rangle = \left\langle \frac{\partial f}{\partial \boldsymbol{u}_s}(\boldsymbol{u}), \boldsymbol{u}_s \right\rangle,$$

$$\sigma_{pq}(\boldsymbol{u}) = \frac{1}{2}\langle \nabla f(\boldsymbol{u}), \nabla j_{pq}(\boldsymbol{u}) \rangle = \frac{1}{2}\left(\left\langle \frac{\partial f}{\partial \boldsymbol{u}_q}(\boldsymbol{u}), \boldsymbol{u}_p \right\rangle + \left\langle \frac{\partial f}{\partial \boldsymbol{u}_p}(\boldsymbol{u}), \boldsymbol{u}_q \right\rangle\right).$$

(39)

*Proof.* We take the definition of the Lagrange multiplier functions in Equation 37. Starting with the denominator, we observe that the Gram matrix is of the form:

$$\mathrm{Gram}(\boldsymbol{u}) = \begin{bmatrix} \boldsymbol{I}_C & \boldsymbol{0} \\ \boldsymbol{0} & 2\boldsymbol{I}_{\frac{C(C-1)}{2}} \end{bmatrix}, \tag{40}$$

since for each block matrix $(\boldsymbol{A}, \boldsymbol{B}, \boldsymbol{C})$ of the Gramian, we get for each of their elements:

- $\forall A_{s,r}$: $\langle \nabla j_s(\boldsymbol{u}), \nabla j_r(\boldsymbol{u}) \rangle = \delta_{sr}$

- $\forall B_{\gamma\tau,\alpha\beta}$: $\langle \nabla j_{\gamma\tau}(\boldsymbol{u}), \nabla j_{\alpha\beta}(\boldsymbol{u}) \rangle = 2\delta_{\gamma\alpha}\delta_{\tau\beta} + 2\delta_{\tau\alpha}\delta_{\gamma\beta}$

- $\forall C_{s,\alpha\beta}$: $\langle \nabla j_s(\boldsymbol{u}), \nabla j_{\alpha\beta}(\boldsymbol{u}) \rangle = 2\delta_{s\alpha}\delta_{s\beta} = 0$

where $\delta_{ij}$ is defined as the Kronecker delta symbol, *i.e.*, $\delta_{ij} = [i = j]$.

Thus, the determinant of this Gram matrix is:

$$\det(\mathrm{Gram}(\boldsymbol{u})) = \det \begin{bmatrix} \boldsymbol{I}_C & \boldsymbol{0} \\ \boldsymbol{0} & 2\boldsymbol{I}_{\frac{C(C-1)}{2}} \end{bmatrix} = \det(\boldsymbol{I}_C) \det\left(2\boldsymbol{I}_{\frac{C(C-1)}{2}}\right) = 2^{\frac{C(C-1)}{2}}. \tag{41}$$

Now, let us turn our focus to the numerator of the Lagrange multiplier function $\sigma_s(\boldsymbol{u})$.

$$\det(\mathrm{Gram}_s(\boldsymbol{u})) = \det \left[ \begin{array}{ccccc|ccc} 1 & \dots & \langle \nabla f(\boldsymbol{u}), \nabla j_1(\boldsymbol{u}) \rangle & \dots & 0 & 0 & \dots & 0 \\ \vdots & \ddots & \vdots & \ddots & \vdots & \vdots & \ddots & \vdots \\ 0 & \dots & \langle \nabla f(\boldsymbol{u}), \nabla j_s(\boldsymbol{u}) \rangle & \dots & 0 & 0 & \dots & 0 \\ \vdots & \ddots & \vdots & \ddots & \vdots & \vdots & \ddots & \vdots \\ 0 & \dots & \langle \nabla f(\boldsymbol{u}), \nabla j_C(\boldsymbol{u}) \rangle & \dots & 1 & 0 & \dots & 0 \\ \hline 0 & \dots & \langle \nabla f(\boldsymbol{u}), \nabla j_{12}(\boldsymbol{u}) \rangle & \dots & 0 & 2 & \dots & 0 \\ \vdots & \ddots & \vdots & \ddots & \vdots & \vdots & \ddots & \vdots \\ 0 & \dots & \langle \nabla f(\boldsymbol{u}), \nabla j_{C-1,C}(\boldsymbol{u}) \rangle & \dots & 0 & 0 & \dots & 2 \end{array} \right]$$

$$= \langle \nabla f(\boldsymbol{u}), \nabla j_s(\boldsymbol{u}) \rangle \cdot \det \begin{bmatrix} \boldsymbol{I}_{C-1} & \boldsymbol{0} \\ \boldsymbol{0} & 2\boldsymbol{I}_{\frac{C(C-1)}{2}} \end{bmatrix}$$

$$= 2^{\frac{C(C-1)}{2}} \langle \nabla f(\boldsymbol{u}), \nabla j_s(\boldsymbol{u}) \rangle. \tag{42}$$

Similarly, for $\sigma_{pq}(\boldsymbol{u})$, we get:

$$\det(\mathrm{Gram}_{pq}(\boldsymbol{u})) = \langle \nabla f(\boldsymbol{u}), \nabla j_{pq}(\boldsymbol{u}) \rangle \cdot \det \begin{bmatrix} \boldsymbol{I}_C & \boldsymbol{0} \\ \boldsymbol{0} & 2\boldsymbol{I}_{\frac{C(C-1)}{2}-1} \end{bmatrix}$$

$$= 2^{\frac{C(C-1)}{2}-1} \langle \nabla f(\boldsymbol{u}), \nabla j_{pq}(\boldsymbol{u}) \rangle. \tag{43}$$

Finally, we get the result by combining the determinants' results for each Lagrange multiplier function and computing the gradients of the constraint functions.

$\square$

Similarly to the gradient vector field, we can show that the Hessian for a constraint manifold (*e.g.*, Stiefel manifold, $\mathrm{Hess}\tilde{f}(\boldsymbol{u}) : T_{\boldsymbol{u}}St_C^d \times T_{\boldsymbol{u}}St_C^d \to \mathbb{R}$) can be given as such [10]:

$$\mathrm{Hess}\tilde{f}(\boldsymbol{u}) = \left( \mathrm{Hess}f(\boldsymbol{u}) - \sum_{1 \le s \le C} \sigma_s(\boldsymbol{u})\mathrm{Hess}j_s(\boldsymbol{u}) \right.$$

$$\left. - \sum_{1 \le p < q \le C} \sigma_{pq}(\boldsymbol{u})\mathrm{Hess}j_{pq}(\boldsymbol{u}) \right)_{|T_{\boldsymbol{u}}St_C^d \times T_{\boldsymbol{u}}St_C^d}. \tag{44}$$

We can transform the results into a matrix form in the following way. First, we denote,

$$\nabla f(\boldsymbol{U}) \triangleq \mathrm{vec}^{-1}(\nabla f(\boldsymbol{u})) \in \mathbb{R}^{d \times C}; \quad \partial f(\boldsymbol{U}) \triangleq \mathrm{vec}^{-1}(\partial f(\boldsymbol{u})) \in \mathbb{R}^{d \times C}. \tag{45}$$

We then introduce a symmetric matrix $\Sigma(\boldsymbol{U}) \triangleq [\sigma_{pq}(\boldsymbol{u})] \in \mathbb{R}^{C \times C}$, where we also include the Lagrange multiplier function's symmetrical component. For the Stiefel manifold, the matrix form of the embedded gradient vector field (after some simple computation) is given by,

$$\partial f(\boldsymbol{U}) = \nabla f(\boldsymbol{U}) - \boldsymbol{U}\Sigma(\boldsymbol{U}), \tag{46}$$

where $\Sigma(\boldsymbol{U}) = \frac{1}{2}\left(\nabla f(\boldsymbol{U})^\top \boldsymbol{U} + \boldsymbol{U}^\top \nabla f(\boldsymbol{U})\right)$.

Regarding the Hessian in Equation 44, to write it in matrix form, we first need to compute the Hessian matrices of the constraint functions as follows[3]:

$$
[\mathrm{Hess}\, j_s(\boldsymbol{U})] = \begin{bmatrix} \boldsymbol{0}_d & \dots & \boldsymbol{0}_d & \dots & \boldsymbol{0}_d \\ \vdots & \ddots & \vdots & \ddots & \vdots \\ \boldsymbol{0}_d & \dots & \boldsymbol{I}_d & \dots & \boldsymbol{0}_d \\ \vdots & \ddots & \vdots & \ddots & \vdots \\ \boldsymbol{0}_d & \dots & \boldsymbol{0}_d & \dots & \boldsymbol{0}_d \end{bmatrix} = \boldsymbol{I}_d \otimes \left(\boldsymbol{e}_s \otimes \boldsymbol{e}_s^\top\right);
$$

$$
[\mathrm{Hess}\, j_{pq}(\boldsymbol{U})] = \begin{bmatrix} \boldsymbol{0}_d & \dots & \boldsymbol{0}_d & \dots & \boldsymbol{0}_d & \dots & \boldsymbol{0}_d \\ \vdots & \ddots & \vdots & \ddots & \vdots & \ddots & \vdots \\ \boldsymbol{0}_d & \dots & \boldsymbol{0}_d & \dots & \boldsymbol{I}_d & \dots & \boldsymbol{0}_d \\ \vdots & \ddots & \vdots & \ddots & \vdots & \ddots & \vdots \\ \boldsymbol{0}_d & \dots & \boldsymbol{I}_d & \dots & \boldsymbol{0}_d & \dots & \boldsymbol{0}_d \\ \vdots & \ddots & \vdots & \ddots & \vdots & \ddots & \vdots \\ \boldsymbol{0}_d & \dots & \boldsymbol{0}_d & \dots & \boldsymbol{0}_d & \dots & \boldsymbol{0}_d \end{bmatrix} = \boldsymbol{I}_d \otimes \left(\boldsymbol{e}_p \otimes \boldsymbol{e}_q^\top + \boldsymbol{e}_q \otimes \boldsymbol{e}_p^\top\right).
$$

(47)

Finally, the matrix form of the Hessian of the cost function $\tilde{f} : St_C^d \to \mathbb{R}$ is given by,

$$
\mathrm{Hess}\, \tilde{f}(\boldsymbol{U}) = \left(\mathrm{Hess}\, f(\boldsymbol{U}) - \boldsymbol{I}_d \otimes \Sigma(\boldsymbol{U})\right)_{|T_{\boldsymbol{U}} St_C^d \times T_{\boldsymbol{U}} St_C^d}, \tag{48}
$$

where from there, we can re-define the expression $\boldsymbol{G}$ in Equation 33 as follows,

$$
\boldsymbol{G} = \mathrm{rvec}(D_{\boldsymbol{U}\boldsymbol{U}}^2 f(\tilde{\boldsymbol{H}}, \boldsymbol{U})) - \boldsymbol{I}_d \otimes \Sigma(\boldsymbol{U}) \in \mathbb{R}^{dC \times dC}. \tag{49}
$$

## C   Additional Experimental Results

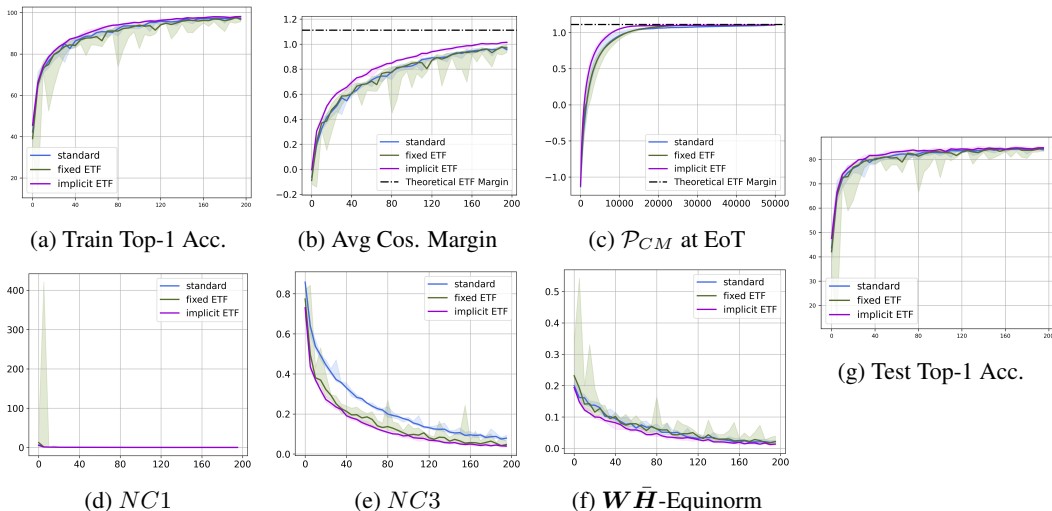

(a) Train Top-1 Acc.    (b) Avg Cos. Margin    (c) $\mathcal{P}_{CM}$ at EoT

(d) $NC1$    (e) $NC3$    (f) $\boldsymbol{W}\bar{\boldsymbol{H}}$-Equinorm

(g) Test Top-1 Acc.

Figure 6: CIFAR10 results on ResNet-18. In all plots, the x-axis represents the number of epochs, except for plot (c), where the x-axis denotes the number of training examples.

---

[3]Here the resulting Kronecker product is expressed in a row-major way.

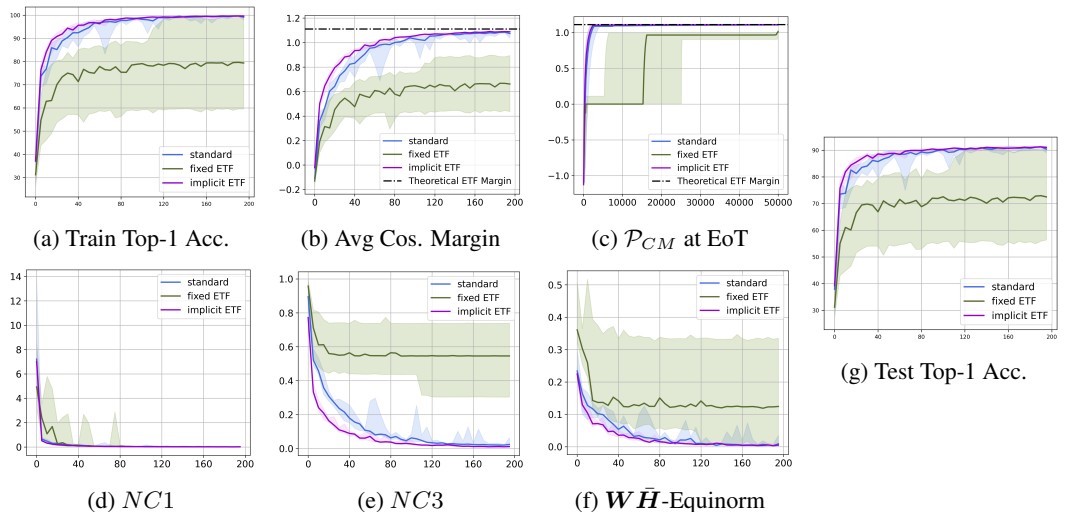

(a) Train Top-1 Acc.  (b) Avg Cos. Margin  (c) $\mathcal{P}_{CM}$ at EoT

(d) $NC1$  (e) $NC3$  (f) $\boldsymbol{W}\bar{\boldsymbol{H}}$-Equinorm

(g) Test Top-1 Acc.

Figure 7: CIFAR10 results on VGG-13. In all plots, the x-axis represents the number of epochs, except for plot (c), where the x-axis denotes the number of training examples.

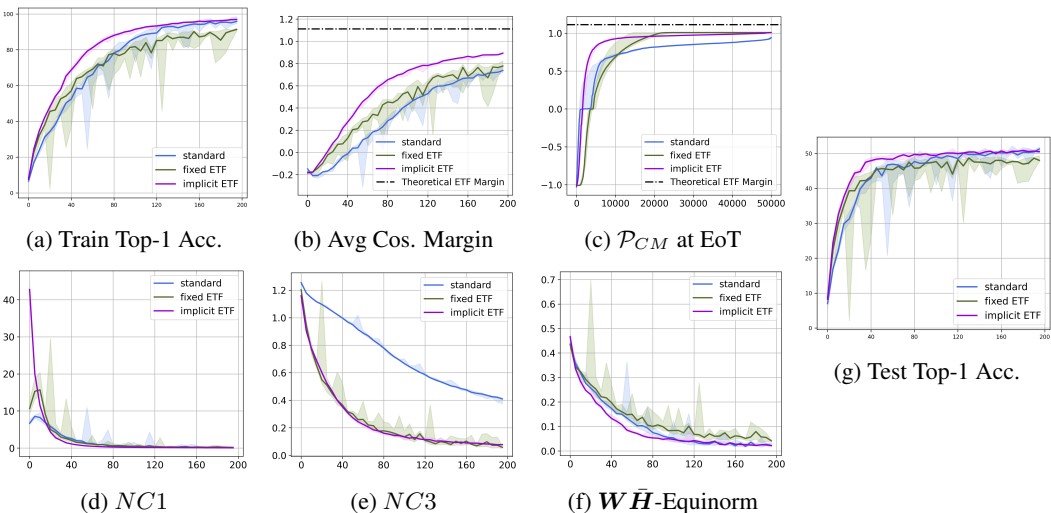

(a) Train Top-1 Acc.  (b) Avg Cos. Margin  (c) $\mathcal{P}_{CM}$ at EoT

(d) $NC1$  (e) $NC3$  (f) $\boldsymbol{W}\bar{\boldsymbol{H}}$-Equinorm

(g) Test Top-1 Acc.

Figure 8: CIFAR100 results on ResNet-50. In all plots, the x-axis represents the number of epochs, except for plot (c), where the x-axis denotes the number of training examples.

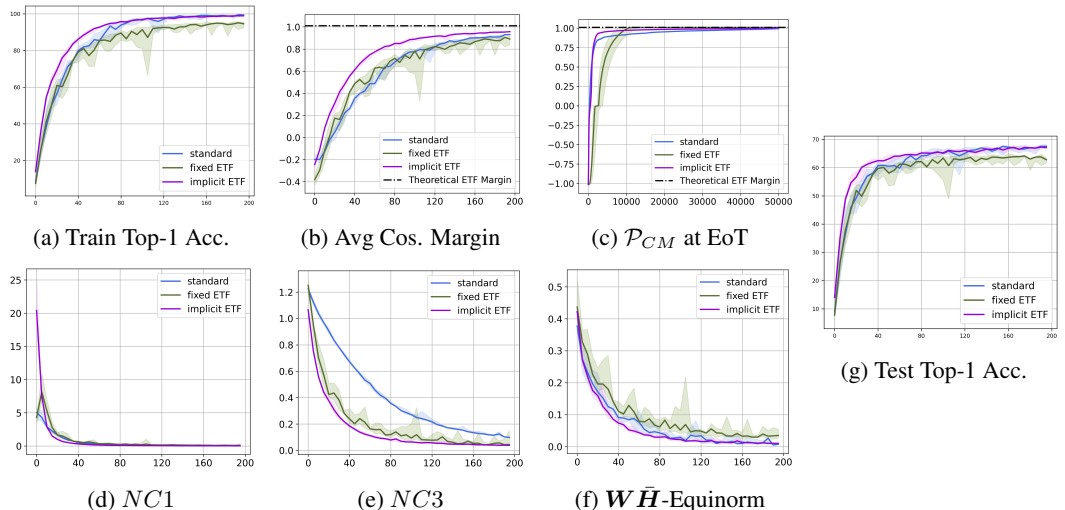

(a) Train Top-1 Acc.  (b) Avg Cos. Margin  (c) $\mathcal{P}_{CM}$ at EoT

(d) $NC1$  (e) $NC3$  (f) $\boldsymbol{W}\bar{\boldsymbol{H}}$-Equinorm

(g) Test Top-1 Acc.

Figure 9: CIFAR100 results on VGG-13. In all plots, the x-axis represents the number of epochs, except for plot (c), where the x-axis denotes the number of training examples.

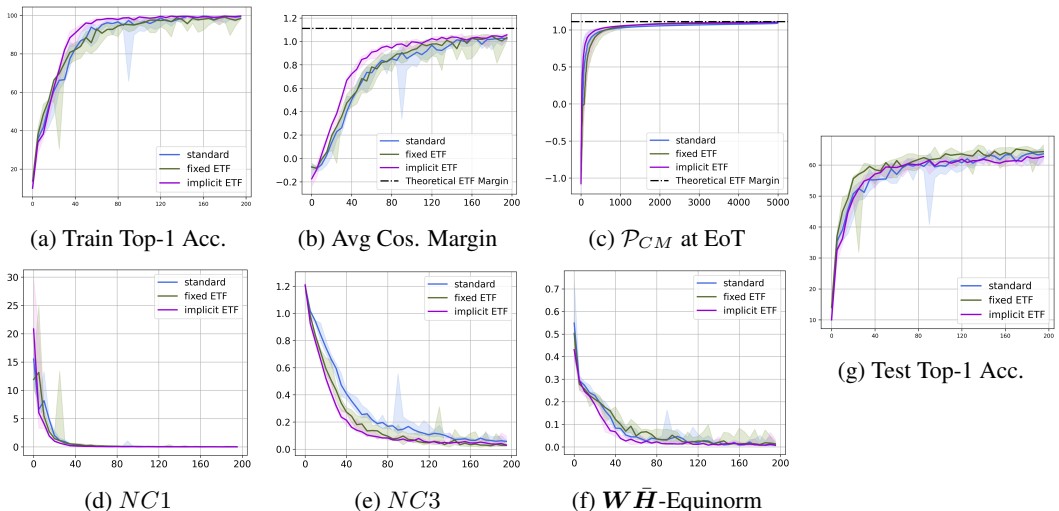

(a) Train Top-1 Acc.  (b) Avg Cos. Margin  (c) $\mathcal{P}_{CM}$ at EoT

(d) $NC1$  (e) $NC3$  (f) $\boldsymbol{W}\bar{\boldsymbol{H}}$-Equinorm

(g) Test Top-1 Acc.

Figure 10: STL10 results on ResNet-50. In all plots, the x-axis represents the number of epochs, except for plot (c), where the x-axis denotes the number of training examples.

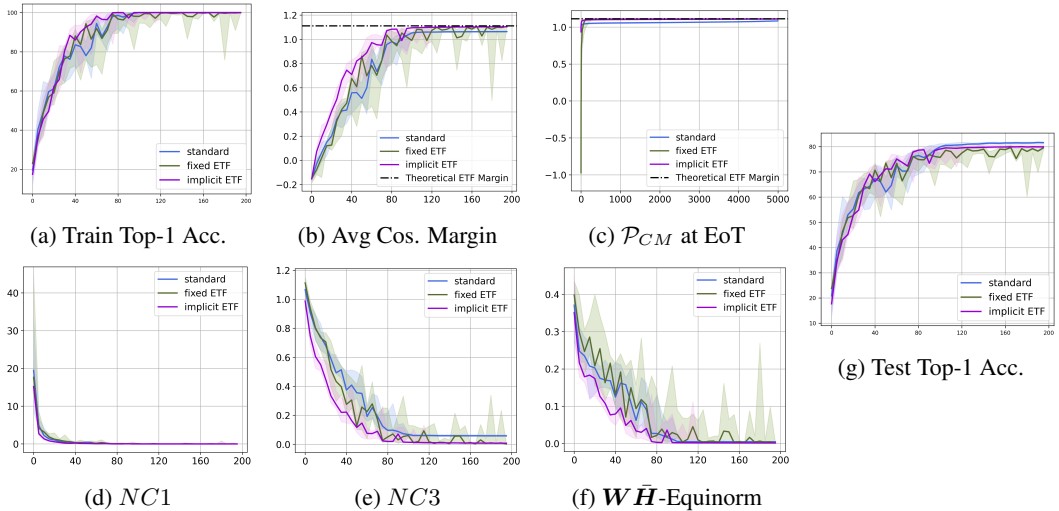

(a) Train Top-1 Acc.     (b) Avg Cos. Margin     (c) $\mathcal{P}_{CM}$ at EoT

(d) $NC1$     (e) $NC3$     (f) $\boldsymbol{W}\bar{\boldsymbol{H}}$-Equinorm

(g) Test Top-1 Acc.

Figure 11: STL10 results on VGG-13. In all plots, the x-axis represents the number of epochs, except for plot (c), where the x-axis denotes the number of training examples.

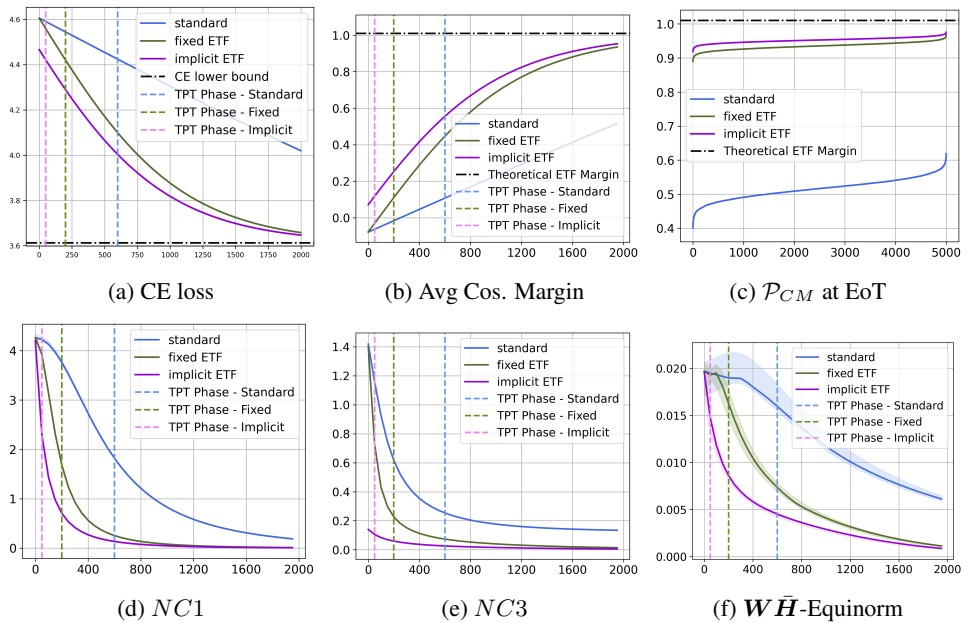

(a) CE loss     (b) Avg Cos. Margin     (c) $\mathcal{P}_{CM}$ at EoT

(d) $NC1$     (e) $NC3$     (f) $\boldsymbol{W}\bar{\boldsymbol{H}}$-Equinorm

Figure 12: UFM-100 results. In all plots, the x-axis represents the number of epochs, except for plot (c), where the x-axis denotes the number of training examples.

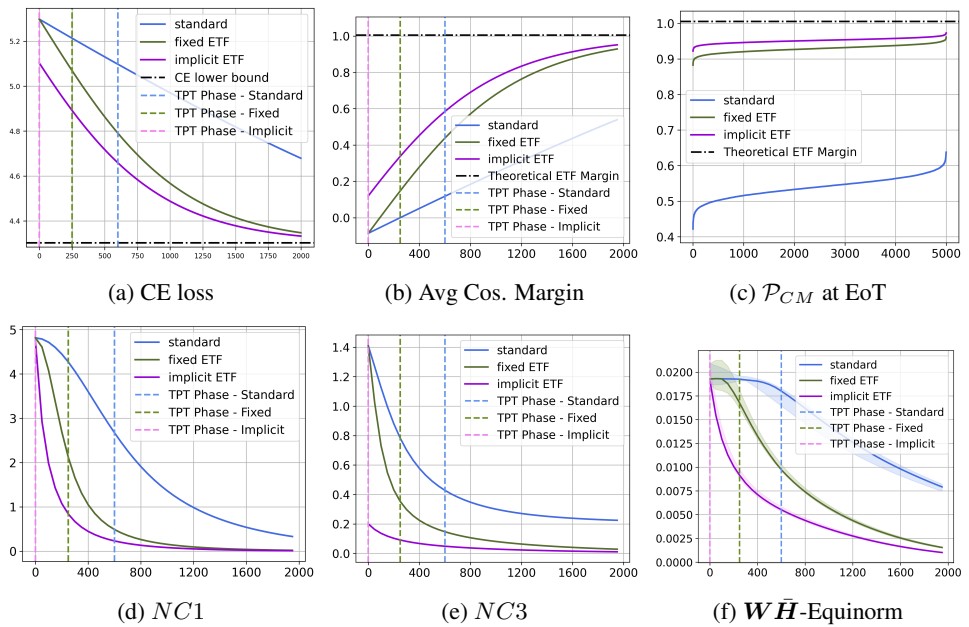

Figure 13: UFM-200 results. In all plots, the x-axis represents the number of epochs, except for plot (c), where the x-axis denotes the number of training examples.

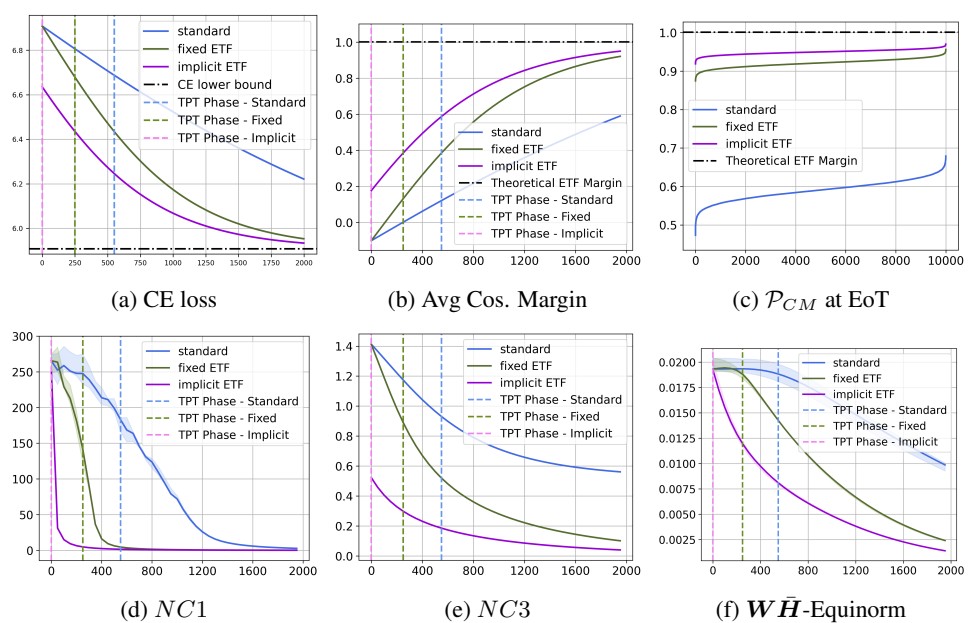

Figure 14: UFM-1000 results. In all plots, the x-axis represents the number of epochs, except for plot (c), where the x-axis denotes the number of training examples.

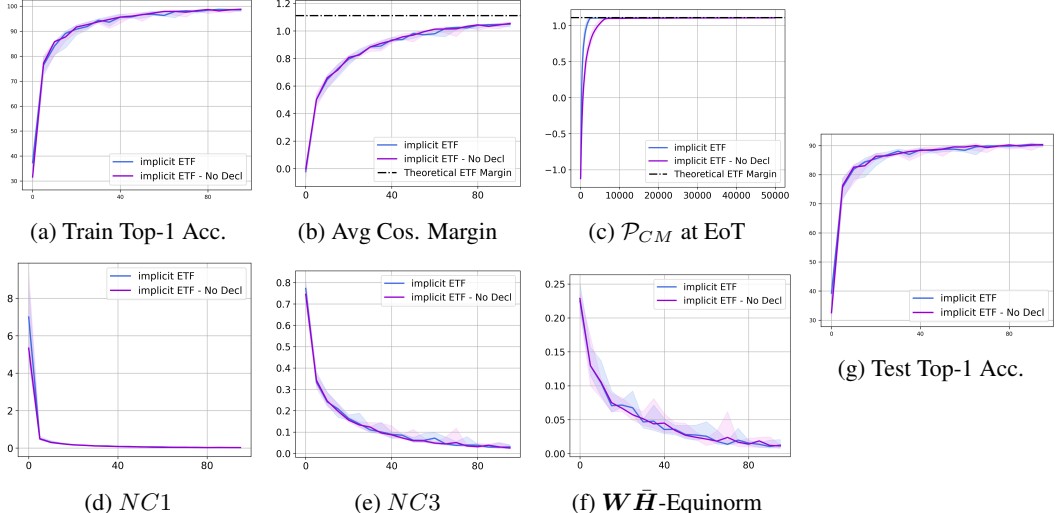

(a) Train Top-1 Acc.   (b) Avg Cos. Margin   (c) $\mathcal{P}_{CM}$ at EoT

(d) $NC1$       (e) $NC3$      (f) $\boldsymbol{W}\bar{\boldsymbol{H}}$-Equinorm

(g) Test Top-1 Acc.

Figure 15: CIFAR10 results on VGG-13, comparing the implicit ETF method in two scenarios: one where the DDN gradient is computed and included in the SGD update, and another where the DDN gradient computation is omitted from the update. In all plots, the x-axis represents the number of epochs, except for plot (c), where the x-axis denotes the number of training examples.

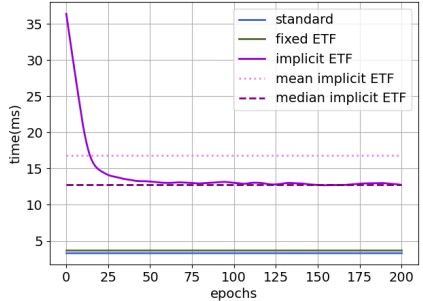

(a) Forward pass times in milliseconds.

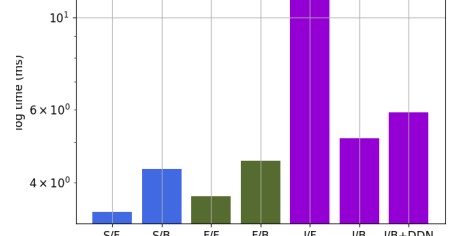

(b) Forward and backward times in (log) millisecs.

Figure 16: CIFAR10 computational cost results on ResNet-18. In (a), we plot the forward pass time for each method. For the implicit ETF method, which has dynamic computation times, we also include the mean and median time values. In (b), we plot the computational cost for each forward and backward pass across methods. For the implicit ETF forward pass, we have taken its median time. The notation is as follows: S/F = Standard Forward Pass, S/B = Standard Backward Pass, F/F = Fixed ETF Forward Pass, F/B = Fixed ETF Backward Pass, I/F = Implicit ETF Forward Pass, and I/B = Implicit ETF Backward Pass.

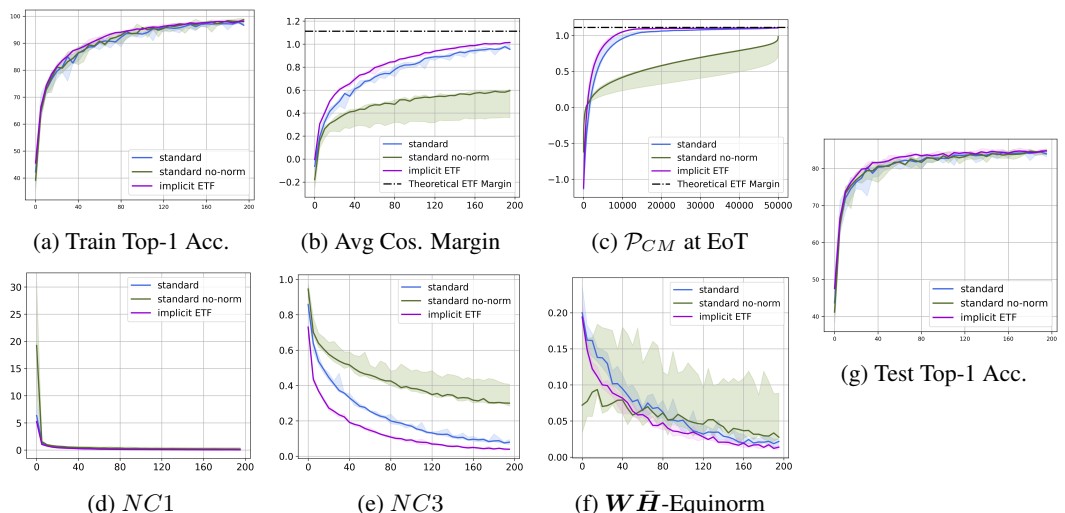

(a) Train Top-1 Acc.    (b) Avg Cos. Margin    (c) $\mathcal{P}_{CM}$ at EoT

(d) $NC1$    (e) $NC3$    (f) $\boldsymbol{W}\bar{\boldsymbol{H}}$-Equinorm

(g) Test Top-1 Acc.

Figure 17: CIFAR10 results on ResNet-18, where in standard no-norm we do not perform feature and weight normalisation. In all plots, the x-axis represents the number of epochs, except for plot (c), where the x-axis denotes the number of training examples.

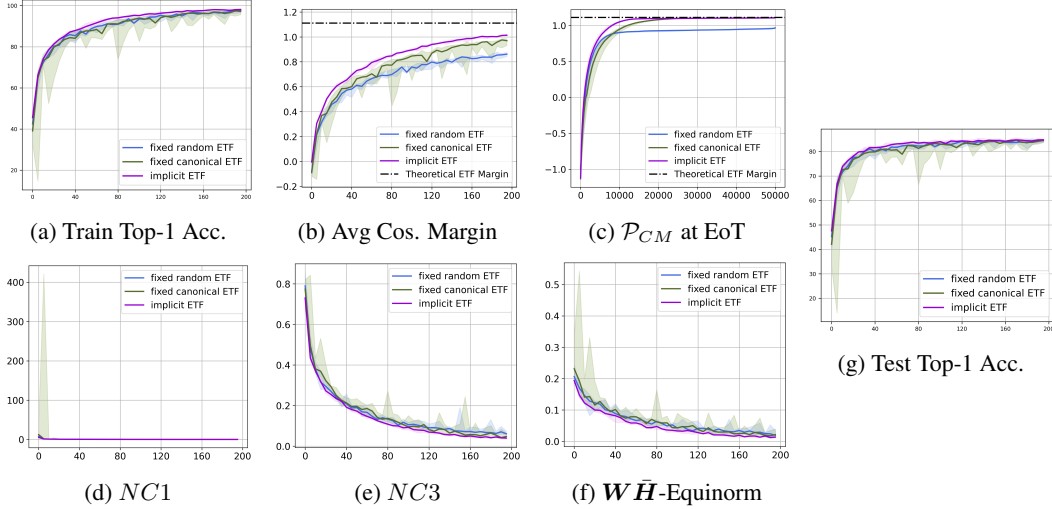

(a) Train Top-1 Acc.    (b) Avg Cos. Margin    (c) $\mathcal{P}_{CM}$ at EoT

(d) $NC1$    (e) $NC3$    (f) $\boldsymbol{W}\bar{\boldsymbol{H}}$-Equinorm

(g) Test Top-1 Acc.

Figure 18: CIFAR10 results on ResNet-18, where in fixed random ETF we choose a random orthogonal direction instead of the canonical one. In all plots, the x-axis represents the number of epochs, except for plot (c), where the x-axis denotes the number of training examples.

