# OpenReview forum: "Guiding Neural Collapse: Optimising Towards the Nearest Simplex Equiangular Tight Frame"
_NeurIPS.cc/2024/Conference — NeurIPS 2024 poster_

### Official Review · Reviewer_isxr · 2024-07-05

**Soundness:** 4
**Presentation:** 4
**Contribution:** 3
**Rating:** 6
**Confidence:** 4

**Summary:**

The paper uses Riemannian optimization to guide the final layer weights (the linear classifier) toward the nearest simplex ETF orientation. In particular, consider the two common approaches of training a deep classifier network:

1. The standard training strategy where the final layer weights are updated by backpropagation.

2. The final layer weights are fixed as a simplex ETF (which has been well-studied in previous works).

The proposed approach leverages the duality between penultimate layer features and the final layer weights (to form a simplex ETF orientation) and gradually guides the latter to an optimal simplex ETF per training step.

**Strengths:**

1. The proposed approach frames the gradual transition of weights to a simplex ETF as a Riemannian optimization problem, which can be differentiated. Thus, allowing for an end-end training pipeline. The combination of these techniques is novel to the neural collapse setting.

2. The experimental results are presented for the simple UFMs as well as practical networks and datasets to showcase the convergence benefits.

**Weaknesses:**

The authors do not provide numerical data for the extra memory and step-time that is required by the extra deep declarative layer. A brief discussion is presented in Section 5 but I believe further details would strengthen the paper. For instance:
- By what percentage does the step time and memory increase when adding this layer?
- When should one avoid the backward pass through this layer and consider only the forward pass?
- What is the dependence on the memory and step time growth with the feature dimension and the number of classes? Maybe a simple UFM-based analysis should suffice.

see more questions below.

**Questions:**

1. How effective is the proposed approach in settings with imbalanced classes [1] ? More generally, for settings where the simplex ETF might not be an ideal configuration (for instance: graph neural networks, see [2] ). A brief discussion on these topics can further strengthen the paper.

2. Instead of running the optimization layer to select the final layer weights at every step, what if we do it after every $k$ step? Can we potentially reduce the majority of the computational overheads while improving the convergence?

3. What is the convergence behavior when employing SGD/Adam instead of the AGD approach?

nit: Where is $U_{init}$ defined?

nit: line 117, below eq (5), is the formulation of $\widetilde{H}$ correct? shouldn't the denominator be $||\overline{H}||_F$ ?

References

[1] Fang, C., He, H., Long, Q., & Su, W. J. (2021). Exploring deep neural networks via layer-peeled model: Minority collapse in imbalanced training. Proceedings of the National Academy of Sciences

[2] Kothapalli, Vignesh, Tom Tirer, and Joan Bruna. "A neural collapse perspective on feature evolution in graph neural networks." Advances in Neural Information Processing Systems 36 (2024).

---

> ### Author Rebuttal · Authors · 2024-08-07
>
> Weaknesses:
>
> Regarding the computational and memory costs of our approach, please refer to our general response.
>
> Avoid backward pass: Theory suggests that incorporating the DDN layer's backward pass should provide additional gradient information for updating the features’ parameters in the backbone neural network. While we have observed that this additional gradient information improves stability, it has not been critical for the algorithm's convergence. Therefore, omitting the backward pass can be a practical choice in situations where computational resources are a concern.
>
> Specifically, in our experiments with ImageNet, the current implementation of the DDN backward pass exceeds GPU memory limits. As a result, we opted not to use it for these experiments. However, we acknowledge the importance of the DDN backward pass for gradients to be technically correct, and plan to address these computational and memory challenges in future work to ensure that it can be effectively utilised even for large-scale datasets such as ImageNet.
>
> Questions:
>
> 1. Minority collapse refers to the training dynamics observed in standard approaches when dealing with significant class imbalance. In such cases, minority classes tend to collapse into a single vertex of a simplex ETF, resulting in a degenerate solution. However, under certain conditions (e.g., when the number of features $d$ exceeds the number of classes $C$), a simplex ETF can still represent an optimal classifier configuration, even in imbalanced settings (see Theorem 1 of Yang et al. [67]). The primary challenge is that a learned classifier may struggle to achieve this configuration easily.
> It’s important to note that the assumption for neural collapse to occur is zero misclassification error during the TPT, which is clearly not met in the case of minority collapse. Substantial research on imbalance training and long-tail classification tasks employs fixed ETF approaches to address problems associated with conventional training methods. We believe that our approach, which induces the nearest ETF direction rather than an arbitrary one, will yield even better and more stable results compared to fixed ETF methods. We have included preliminary results (found in the attached pdf) on CIFAR100LT with an imbalanced ratio of 0.01 (following Yang et al. [65]) using a VGG network to support our claims. We have not considered graph neural networks in this work. Our method applies to the structure of the classifier solution as defined by the loss function. Exploration of alternative architectures and loss functions is left for future work.
>
> 2. Running the Riemannian optimisation problem every $k$ steps is indeed a viable extension of our method and can potentially reduce computational demands. We conducted experiments where $k$ was set to the number of steps per epoch. Our findings indicate that optimising once per epoch is insufficient for guiding the network to the nearest ETF and achieving rapid convergence, as the features can change significantly within an epoch. Thus, we opted to perform the optimisation at every step. Hence, we present a trade-off between achieving faster and more stable convergence versus higher computational cost. Introducing $k>1$ as an additional hyperparameter requires careful tuning for each specific case.
> Another advantage of our approach is that, as training progresses, the target ETF stabilises because the features approach convergence. At this point, the Riemannian optimisation problem yields similar solutions. To eliminate the additional computational cost, we can fix the simplex ETF determined by the Riemannian optimisation and continue to converge towards this fixed ETF. However, determining the exact epoch at which this stability is achieved remains heuristic and lacks theoretical guarantees, thus requiring case-by-case consideration.
>
> 3. Our approach is particularly well-suited for optimisers that dynamically adjust step sizes based on the current model state, such as AGD. In contrast, using methods like SGD or Adam with our approach requires careful tuning of the learning rate. This may involve developing a new type of scheduling that considers the objective value of the Riemannian optimisation problem to ensure that the learning rate is appropriately decayed as the solution approaches the ETF. Current learning rate schedulers are typically derived heuristically and optimised for conventional training scenarios, which may not account for the specific needs of our approach. Therefore, adapting these schedulers for our approach requires additional analysis, which is beyond the scope of this paper and will be addressed in future work.
>
> Nit: $U_{init}$ is defined in the paragraph for Hyperparameter Selection and Riemannian Initialisation Schemes in lines 230-235 in our paper.
>
> Nit: Thank you. This is a typo. The Frobenius norm should be taken wrt the feature means and not the features, so it should be $||\bar{H}||_F$.

---

> > ### Comment · Reviewer_isxr · 2024-08-09
> >
> > Thank you for answering the questions.
> >
> > Based on the memory and compute overheads presented in Table 1, seems like the implicit ETF approach is quite slow in terms of step times. For instance, for CIFAR100 / Res50	the implicit ETF approach (fwd+bwd w/o DDN) seems to be ~4-5x slower than the standard fwd + bwd. This factor is much larger for bwd + w/ DDN.
> >
> > **Suggestion:** Taking a step back, since we care about the terminal phase of training for NC analysis, a much better comparison would be to consider `num_steps * step_time` for each of the approaches (standard/implicit ETF/explicit ETF). This way, one can know how fast they can converge to TPT within a given time budget.
> >
> > Please clarify these aspects in your claims and consider incorporating a discussion of such training efficiency aspects. Good luck.

---

> > > ### Author Response · Authors · 2024-08-10
> > >
> > > Thank you for your suggestions. We will expand our discussion to include these concepts and experiments in our paper.
> > >
> > > In particular, we will provide a more nuanced analysis than the averages reported in Table 1. As training progresses, the Riemannian optimisation (to obtain the implicit ETF in the forward pass) converges more quickly. Indeed, the majority of the cost comes from the initial few iterations.
> > >
> > > To provide a more comprehensive view, we present additional time measurements from a new run of CIFAR-100 using ResNet-50:
> > >
> > >  - Average forward time: 74 ms
> > >  - Median forward time: 17.6 ms
> > >  - Maximum forward time: 825 ms
> > >  - Minimum forward time: 14 ms
> > >
> > > These results indicate that the median forward time is competitive with other methods. The significant variance in forward time (from a minimum of 14 ms to a maximum of 825 ms) highlights the variable nature of the forward pass. However, the median value suggests that the majority of forward passes are performed efficiently, reinforcing the overall competitiveness of our approach. This trend is consistent across other datasets and architectures as well.
> > >
> > > In practice, one way to mitigate the initial overhead is to use a warm-up phase. During this phase, we can apply the standard method for a few iterations to update the features before transitioning to our approach and Riemannian optimisation. However, such practical considerations complicate the presentation of this paper and will be explored in future work.

---

### Official Review · Reviewer_oHwR · 2024-07-09

**Soundness:** 3
**Presentation:** 2
**Contribution:** 3
**Rating:** 6
**Confidence:** 3

**Summary:**

This paper proposed a novel algorithm for neural network training. The algorithm is motivated by the recent discovery on the neural collapse phenomenon, which demonstrates that the last layer of neural network classifier will converge to a specific structure named simplex ETF. The authors propose to guide the network parameters to the ETF structure via explicitly penalizing on the distance to the ETF, and further address the non-uniqueness of the solution via adding a proximal term. Experimental results on various neural network architecture and real world datasets are presented, and the proposed algorithm can universally improve the training and testing accuracy over the standard training.

**Strengths:**

The proposed algorithm is novel and well motivated, and it shows universal and significant improvement over multiple choices of network architecture and datasets. The contribution of this work is solid, it helps the community to understand the benefit of the neural collapse phenomenon, and can potentially improve the standard paradigm of neural network training.

**Weaknesses:**

1. The presentation should be improved, see questions for detail. In general the authors should give more detailed information about how the algorithm is implemented.

2. Although the accuracy on train and test dataset exhibits significant improvement within the fixed number of training epochs, the proposed algorithm are much more complicated to compute. Therefore it makes more sense to compare the running time and computational cost with Standard and Fixed ETF.

3. Proper ablation study is missing. The authors add many additional techniques, such as exponential moving average, stratified batch sampling, deep declarative layer to improve the training. It is not clear how much the improvement indeed comes from the nearest ETF optimization.

**Questions:**

1. How is equation 8 being optimized? Is it using Lagrangian multiplier method? A pseudo code of the proposed algorithm will be very helpful.

2. I found the Proposition 1 hard to follow. The authors should explain what is the implication of this proposition and how does it helps to the stability of training. The current statement is confusing to distinguish the main result, and notations such as $D_y$ and $\Lambda$ are not properly introduced. This proposition should be improved.

3. In Table 1 and 2, fixed ETF on CIFAR10 with VGG seems to have much worse performance than others. Do you have insights about what is going on here?

**Limitations:**

The authors have discussed the limitation properly in the paper. A more detailed discussion with empirical results on the computational cost will be helpful.

---

> ### Author Rebuttal · Authors · 2024-08-07
>
> Weakness 1 with question 1 & 2:
>
> Eq. 8 is optimised as a bi-level optimisation problem. At each gradient update step, we first solve the inner optimisation problem to obtain the nearest ETF solution from the Riemannian problem. This gives the classifier weights directly. Subsequently, we perform the outer optimisation by optimising the rest of the network using stochastic gradient descent.
>
> Proposition 1 restates a known result for gradient computation in differentiable optimisation problems, utilising the implicit function theorem for implicit differentiation (Gould et al. [21], Figure 2 and Section 4.1 provide the underlying intuition).
>
> As illustrated in Figure 1 of our paper, two streams of information contribute to the final loss function. Proposition 1 enables us to backpropagate through both streams of information, theoretically enhancing the stability of our solution. The key insight is that incorporating the second stream of gradient information from the Riemannian optimisation allows us to account for changes in the features $H$ and adjust the network’s parameters accordingly. In practice, we have observed that the gradient magnitude of the second stream (bottom in Figure 1) is relatively small compared to the first stream (top in Figure 1). Coupled with the additional computational cost of computing the DDN gradients, this contribution is somewhat restricted in this case.
>
> Weakness 2: Please refer to our general response for computational cost discussions.
>
> Weakness 3:
>
> Ablation studies: Our goal is to find the nearest simplex ETF of the feature means while considering all the features from the network. In the UFM case, where full-batch gradient updates are performed, all features are available, so additional techniques such as exponential moving averages are not necessary. However, in real-world scenarios with stochastic updates, optimising towards the nearest ETF for the features in a given batch can lead to highly variable results, as we do not have the full picture, and the feature means are constantly changing. By incorporating the described techniques, we ensure that we are moving towards a stable ETF target, accounting only for changes in feature weights after each gradient step.
>
> Question 3:
>
> Fixed ETF results: During VGG training with the fixed ETF case, we observed significant variability in the solutions, with CIFAR-10 showing the largest variability. This variation highlights the impact that fixing to a predefined ETF direction can have on the quality of the solution space. While ResNets, with their residual connections, can somewhat mitigate this issue, VGG-type networks appear to struggle more with fixed ETF directions. The intuition is that depending on the initialisation seed, we may start closer to or further from the chosen ETF solution, leading to results where fixed ETF performs either comparably or inferiorly.
> In contrast to fixed ETFs, the standard approach can sometimes be more robust and perform better. However, this comes with increased memory and computational costs due to the need to learn the classifier. This is because, with the standard method, the classifier has the flexibility to adapt and better match the features, eventually reaching a neural collapse solution in practice. Our approach, on the other hand, is robust to initialisation seeds by moving towards the closest ETF solution from any starting point and achieves such convergence more quickly.

---

> > ### Comment · Reviewer_oHwR · 2024-08-10
> >
> > Thanks for the response. I have no further concerns and would like to keep my score at this moment. I encourage the authors to explore further how to improve computational efficiency, and I believe it will lead to a solid contribution to the community.

---

### Official Review · Reviewer_m1UJ · 2024-07-12

**Soundness:** 3
**Presentation:** 3
**Contribution:** 3
**Rating:** 6
**Confidence:** 3

**Summary:**

One of the key aspects of neural collapse (NC) is that the penultimate class feature means form a simplex Equiangular Tight Frame (ETF). The main idea of this paper is to leverage this insight and improve training by further encouraging this property during training. The authors suggest doing this by solving a Riemannian optimization at a given iteration. The way it works is that the classifier weights are set to the nearest simplex ETF obtained by solving this inner Riemannian optimization problem. The classifier weights are dynamically updated during training using this Riemannian opitmization problem at each iteration (rather than trained using gradient descent) using a "deep declarative node" this allows gradients to propagate through the Riemannian optimization.

They show that this approach indeed speeds up training convergence and improves training stability. Their experiments include both synthetic and real-world classification tasks and architectures.

Overall the authors present a nice idea and it is a well-written paper. However, there are a few issues related to the experiments that I outline below.

From my viewpoint, the value of this paper and their method (to me) is less the improved test accuracy and more the improved stability and speed of convergence. It's important to note that this speed up also comes at an additional cost (i.e. in performing the Riemannian optimization). Therefore, the improvements to stability or speed of convergences should be weighted against this caveat. I think it would help to highlight this tradeoff more upfront and make that more clear/transparent.

**Strengths:**

This is a thoughtful and well-written paper. The authors suggest a nice idea to leverage this insight of NC in deep learning and their approach has clear benefits. It is a nice idea and very well executed.

There are clear improvements to the current methods; e.g., their improvement upon [74] by solving the inner Riemannian optimization instead of requiring the model backbone to do the work of matching to a chosen fixed ETF.

The theory and the idea is very compelling. The implementation is good and well explained. Beyond the theory and the novelty of the idea, the main strength of the paper is the value added wrt convergence speed in terms of the number of epochs required for the network to converge.

Good work.

**Weaknesses:**

The main points of concern for me are in regards to the experiments and how the results are reported in the paper.

Table 2 looks good but is a bit misleading particularly when comparing the ranges of the test top-1 accuracy.
The results are still interesting but it's not such a strong/clear winner; that is, when looking at the ranges, it's not so obvious. The authors point this out and clarify that the advantages are speed to convergence and decreased variability which I agree are definite plusses.

The test top-1 accuracies reported in Table 2 aren't competitive with what can be obtained on these benchmark datasets, particularly for the Resnet models. For example, looking at 200 epochs or training, STL on ResNet50 should be able to achieve 85-90% test accuracy, even for Resnet18 the test top-1 accuracy for STL should be upwards of 75%. Similarly, for CIFAR100 on Resnet50, the test accuracies aren't competitive. It'd be interesting to see if these claims about variability still hold when giving the baselines adequate chance to be competitive.

For Figure 4, also no error bars. Understanding compute restraints, it would be nice to see similar multiple seed runs for ImageNet experiments.

Finally, one thing that is not reported here is an estimate of compute cost. Their method requires additional compute for each iteration. Perhaps when compared on this axis their implicit ETF and the Standard training method would be more fairly compared.
The authors do mention this in the limitations section.

**Questions:**

How do you know that the ETF that you steer towards via this Riemannian optimization process is better than the one that you would have arrived at naturally? You say "this process [provides] the algorithm with a starting point that is closer to an optimal solution rather than requiring it to learn a simplex ETF or converge towards an arbitrary one". How do you know that this is optimal? Optimal in what context? If I understand correctly, it's just the solution of the Riemannian optimization which means it forces the class means into an ETF. It's optimal wrt to the optimizaiton problem but not necessarily for the learning task? Is that correct?

Do you do any, or is it possible to perform a comparison of these two resulting ETFs?
How does the test accuracy of your 'encouraged' ETF compare to the one you would have obtained naturally?

In Section 3.3. The Proximal Problem. I just don't see immediately why adding the proximal term guarantees uniqueness of the solution and how it stabilizes the Riemannian optimization problem. Can you add more detail or proof or reference to proof?

On first reading, it was unclear to me exactly how is U_prox defined? And what is used in practice. Is it determined from the previous iteration? If I understand correctly, you tried two approaches: setting U_init = U_prox = canonical ETF. Or to set both equal to random orthogonal matrices from classical compact groups.
It sounds like, in the end, you run training without the proximal matrix for one epoch. Then use the resulting U* to set U_init = U_prox = U* from that one epoch. Is that correct? How was this "warmup" approach validated? Did you experiment with various epochs? How stable were the outcomes of that analysis? You later mention (line 225) that the correct choice of these values is "crucial" so it seems important to understand.

In the Section Hyperparameter Selection and Riemannian Initialization Schemes: You mention that algorithm convergence is robust to values of \delta and that the \delta reg term is a trade-off between the optimal solution's proximity to the feature means and its proximity to the given simplex ETF direction. Did you explore how and when to introduce this constraint? Or any exploration of how the solution varies with \delta?

In section 3.4 General learning Setting: The role of the temperature \tau is bit unclear to me. And the reference to [67, Theorem 1] isn't very helpful. Perhaps a little more clarity as to the role \tau plays here? You state later in the Experiments section  that you use \tau=5 according to Yaras et al [67]. This hyperparam choice is not very clear to me.

(typo? clarification?) Proposition 1. There is notation discrepancy between what is stated in the Proposition and what is derived in the Appendix B. Namely, the Proposition is stated wrt \bar{H} but the derivation is carried out for \tilde{H}. I understand that \tilde{H} is the normalized (wrt Frobenius) matrix \bar{H} so perhaps it all works out with the normalization constant but the discrepancy there and comparing back with dimensionality of matrices in the original statement of Proposition 4.5 in Gould et al. [21] (from which this result follows) had me a bit confused.

Are there error bars in Figure 2? I see them for plot (f) but not for the others?
(clarification) What is depicted in Figure 2(c)? What is \mathcal{P}_{CM}? I think I somehow missed that.

Are there error bars in Figure 3? Were multiple trials run for these experiments?

Tables 1 and 2: The ranges for train and test top-1 accuracy values for STL on VGG seem very large.

In regards to Figure 4, I'd recommend either performing more training runs for Imagenet on Resnet50. The results look very compelling but without error bars don't say much. Similarly, comparing the results in Figure 4 with those for the other real-world datasets (e.g. Cifar10, Cifar100, STL) those contained in the Appendix which do have error bars are arguably less convincing of the primarily claims of speed to convergence.

**Limitations:**

N/A The authors address any limitations.

---

> ### Author Rebuttal · Authors · 2024-08-07
>
> Weaknesses:
>
> Misleading results: The reviewer correctly observed that, particularly on smaller datasets, the performance converges to be approximately equivalent by the end of training. Any observed deviations are likely due to random effects. This is to be expected, as all properly trained ETF solutions (i.e., with non-random labels) should theoretically yield similar results. Our work leverages the symmetries inherent in the optimal solution geometry to reach an ETF solution in fewer iterations. Thus, the primary advantage of our approach lies in its faster convergence while maintaining comparable generalisation ability. Notably, in some cases, our method demonstrates superior generalisation performance because, within the given training time, other approaches may have yet to reach the optimal max-margin solution.
>
> Non-competitive results: In our experiments, we utilised the AGD optimiser. While AGD achieves SoTA results for VGG architectures, it is not yet fully tuned for ResNet architectures. However, it has been demonstrated that state-of-the-art results can be obtained for ResNet with prolonged training (see [6]). Our focus, however, is not on achieving state-of-the-art results but rather on illustrating the convergence trajectory. Due to computational constraints and the required scale of experiments, we terminated training at 200 epochs.
>
> Figure 4 no error bars: Fixed and replaced with 5 seed runs (found in the attached pdf).
>
> Cost: Please see tables and discussion in the general response.
>
> Questions:
>
> Optimal ETF solution: As previously discussed, all ETF solutions are equivalent up to rotations and permutations. Our Riemannian optimisation approach ensures that with each gradient update step, we move towards the nearest ETF solution, effectively breaking these symmetries and achieving faster convergence. Consequently, while our solution is optimal wrt the Riemannian problem, it is also optimal for the learning problem, although no better than any other ETF solution.
>
> Compare two ETFs (ours vs fixed): In theory, we expect the testing accuracy of the two solutions to be equivalent. However, in practice, due to the finite amount of training and the complexity of tasks such as those with many classes (e.g., ImageNet), we observe that faster convergence to the ETF solution can lead to better generalisation performance. However, we are careful not to make any theoretical claims in this regard.
>
> Proximal term: Solving the original problem in Eq. 6 yields a non-unique solution due to the rank deficiency of matrix $M$, resulting in a family of possible solutions. To achieve a unique solution, we introduce a proximal term to make the problem strongly convex, as shown in Eq. 7. This approach stabilises the solution. Additionally, the DDN backward pass for Eq. 6 cannot be computed deterministically because of the singularity of the gradients. By incorporating the proximal term, we obtain non-singular gradients, which allows us to compute Prop. 1.
>
> $U_{prox}$: An obvious initialisation scheme for both $U_{prox}$ and $U_{init}$​ is to set them as either canonical or randomly orthogonal directions. However, both approaches are sensitive to the initial parameter settings of the network. We found that the most effective strategy is to solve the Riemannian problem in Eq. 6 at the first gradient update step (corresponding to the first epoch in GD and the first mini-batch in SGD) to obtain a solution from the family of solutions. This solution, denoted $U*$, is then used as the initialisation for both $U_{init}$ and $U_{prox}$ in Eq. 7.  We then solve Eq. 7 and perform the first backward pass. The intuition behind this initialisation scheme is to start with a solution that is both feasible and optimal for the original problem formulation. Empirically, this approach has yielded the best and most stable results.
>
> $\delta$ proximal param: $\delta$ is a tradeoff between the distance of the feature means from an ETF and the distance between the target direction ($U_{prox}$) and the currently optimised direction. We have found that a broad range of $\delta$ values provides stable and good performance. Therefore, $\delta$ can be set within this range, provided it is not too small (e.g., $\delta\ll 10^{-7}$, which essentially solves Eq. 6) and not too large (e.g., $\delta\gg 10$, where the proximal term dominates and results in a fixed ETF case).
>
> $\tau$ param in CE: Since we normalise features, the cross-entropy loss will have a non-zero lower bound (Thm. 1 of Yaras et al. [67]). $\tau$ adjusts this lower bound, bringing it closer to zero. According to Yaras et al., including this parameter results in more defined NC metrics while keeping accuracy unchanged. They empirically found that setting $\tau=5$ yields optimal results for real-world datasets and architectures, and we adopt this approach in our work.
>
> Clarification on Prop.1: The results in the proposition are currently presented wrt the unnormalised feature means matrix $\bar{H}$. However, this choice does not affect the result or its derivation. For consistency, we will update the proposition to use the normalised values, i.e., $\tilde{H}$. Gould et al. [21] derived their result for vector functions with vector variables, whereas we have generalised this result to matrix functions and matrix variables as a natural extension.
>
> Error bars in Figures: All experiments were conducted five times.  The results for UFM (Fig 2, 3) are highly stable, with minimal deviation, and the ranges are easily visible due to the plots' linewidth. $P_{CM}$, defined in Eq. 16, represents the cosine margin of each specific feature.
>
> STL VGG range: We applied our method to various datasets and architectures without case-by-case tuning. We suspect that the initial variation observed in the STL case is related to the relatively large batch size used for that dataset. However, our method effectively reduces the variance within just a few epochs compared to the fixed ETF.

---

> > ### Comment · Reviewer_m1UJ · 2024-08-13
> >
> > Thank you for the clarifications to the questions and comments raised. I agree one of the primary advantages of your approach is in the increased speed to convergence. In this regard, this seems to be a useful technique. It also helps to have more clear insight in the computation cost incurred. I believe clarifying these additional points can help with the message the paper. I will maintain my score is it is.

---

### Official Review · Reviewer_xmab · 2024-07-12

**Soundness:** 3
**Presentation:** 3
**Contribution:** 2
**Rating:** 5
**Confidence:** 4

**Summary:**

This paper presents a novel approach to utilizing ETF geometry. Instead of fixing the weights or making them learnable, their approach dynamically adjusts the weights by solving a Riemannian optimization problem while allowing end-to-end training. They show that their approach outperforms both the fixed and learnable approaches in terms of convergence speed and generalization.

**Strengths:**

Originality:
The idea of dynamically adjusting weights is not new, but in the context of neural collapse (NC), it is a natural extension. Fully learnable weights do not provide the ETF structure, and fixed weights are too restrictive. The proposed approach is a good compromise between the two and combines the best of both worlds.

Quality:
The paper is well-written, and the proposed approach is carefully supported by theorems and experiments.

Clarity:
The paper is well-written and easy to follow.

Significance:
Their approach is general and could be applied to a range of problems. The authors applied it to synthetic UFMs and some standard image benchmarks (CIFAR-10, CIFAR-100, STL-10, ImageNet). The authors plan to release code upon acceptance.

**Weaknesses:**

Overhead Cost:
The proposed method computes the exponential moving average of the feature means, performs a Riemannian optimization, and computes the gradient of DDN. These components introduce overhead in terms of epoch time. The authors claimed in the paper that the gradient of DDN is not computed, and the Riemannian optimization overhead is negligible. This unsupported claim should be backed up by an additional experiment that reports these extra computation times.

Standard Procedure:
"To ensure fair method comparison," the authors include classifier weight normalization and feature normalization for the standard procedure. This is usually not the case when using CE loss (see Fig 2). The authors should justify this choice by providing the results without these normalizations for the standard procedure.

Image Baselines Results are not SOTA:
The reported results are not state-of-the-art. For example, ResNet-18 trained on CIFAR-10 only reaches 80.47%. It seems that these baselines are not well-tuned, and the gain of the proposed approach is not clear and could potentially fade away with a better-tuned baseline. Can the authors comment on this? Additionally, the authors should include the results using ResNet-50 on ImageNet, which should provide a stronger reference point.

Fixed ETF Procedure:
The authors only used the canonical simplex ETF for the fixed procedure. The weight matrix results in many zeros and could lead to poor performance when used as the fixed classifier because some neurons will be inactive. The authors should include the results using the fixed ETF with a non-canonical (i.e., projection on a random basis).

Remarks:
The authors should directly clarify in Tables 1 and 2 the ResNet architecture used (18 or 50).

**Questions:**

See weaknesses section.

**Limitations:**

For large-scale problems, the computational cost of the proposed approach could be a limitation due to the high memory cost of computing the backward pass of the Riemannian optimization. Therefore, the authors did not compute the gradient calculation when reporting their results for the image benchmarks.
The authors claimed that they empirically observed no significant difference in performance for small-scale problems where the DDN gradient can be computed.
Including these results in the supplementary material would also be beneficial.
Moreover, we agree that future work should verify if this is true for large-scale problems.

---

> ### Author Rebuttal · Authors · 2024-08-07
>
> Overhead Cost: Please refer to the tables in our general response and the discussion around computational concerns.
>
> Standard Procedure: Our new experiments show that the standard method, which excludes feature and weight normalisation, achieves similar performance (found in the attached pdf). However, we observe that the learned features are less well-aligned with a max-margin classifier compared to when normalisation schemes are included.
>
> Image Baselines are not SOTA: Since our results overlap with those previously reported in the AGD paper, it is demonstrated that state-of-the-art results can be achieved by extending training over more epochs using the AGD optimiser (Figure 4 in Berstein et al. (2023)). However, the primary focus of our work is not to establish a new state-of-the-art result but to study the neural collapse phenomenon. We originally included results using ResNet50 on ImageNet (Figure 4 in our paper), and in the attached pdf, we also included ImageNet results after five runs with error bars.
>
> Fixed ETF procedure: The reviewer’s observation that the canonical projection results in a sparse weight matrix would have been true if we were working with canonical orthogonal frames. For canonical simplex equiangular tight frames, the sparsity is observed in the semi-orthogonal matrix  $U$ (i.e., $U$ is the identity matrix), not in the weight matrix itself, which remains dense (as shown in Equation 1). To address this concern, we conducted additional experiments with the fixed method on CIFAR-10 using ResNet-18, where we replaced the canonical rotation with a random orthogonal rotation (drawn from the Haar measure). Our results indicate that the performance remains superior and that the fixed and canonical methods yield nearly identical outcomes (found in the attached pdf).
>
> Remark: We have updated our table results to reflect the chosen ResNet and VGG architectures.
>
> Limitations:
>
> We will include the results in the supplementary material that test whether computing the DDN gradient affects the stability of our solution on small-scale problems.

---

### Author Rebuttal · Authors · 2024-08-07

We thank all the reviewers for their time and thoughtful feedback. Here we address the common question regarding the computational costs of our method, and we will address individual comments for each reviewer separately.


Please refer to Table 1 for computational cost and Table 2 for memory cost measurements. The key insight is that incorporating DDN into the backward pass (in our Python implementation) becomes prohibitively expensive when significantly increasing the number of penultimate layer features and classes. However, as demonstrated in the ImageNet case, where GPU memory constraints prevent us from running this step, omitting the DDN backward pass (similar to a gradient stop operation) still yields strong results as we are still directing the solution to the nearest ETF. Note that there is still an indirect gradient path via the classifier, as shown in Figure 1 of our paper. Despite the strong results, we acknowledge the computational challenges associated with the DDN backward pass. The primary bottleneck in GPU memory usage is during the calculation of the constrained derivatives (Eq. 29), where explicit formulation of certain intermediate matrices occurs. Concerning the time complexity of the DDN backward pass, we use a direct solver to compute the expression in Proposition 1, which can be inefficient for large-scale linear systems. We plan to address these limitations (both time and memory) by using more efficient linear algebra solvers in future work.


                               Table 1: Average time (in milliseconds) of a step update during training

| Model                | Standard Fwd | Standard Bwd | Fixed ETF Fwd | Fixed ETF Bwd | Implicit ETF Fwd | Implicit ETF Bwd (w/o DDN) | Implicit ETF Bwd (w/ DDN) |
|----------------------|--------------|--------------|---------------|---------------|------------------|----------------------------|---------------------------|
| UFM-10               | 0.2          | 0.4          | 0.2           | 0.4           | 7.6              | 1.2                        | 1.8                       |
| UFM-100              | 0.2          | 1.0          | 0.4           | 0.8           | 8.5              | 1.1                        | 608.9                     |
| UFM-200              | 0.3          | 1.1          | 0.3           | 0.9           | 8.7              | 1.2                        | 17,909.9                  |
| UFM-1000             | 0.3          | 1.1          | 0.5           | 1.1           | 23.1             | 0.8                        | N/A                       |
| CIFAR10 / Res18      | 3.4          | 4.3          | 3.7           | 4.5           | 12.8             | 5.1                        | 5.9                       |
| CIFAR10 / Res50      | 7.0          | 9.2          | 6.9           | 7.9           | 19.4             | 9.2                        | 15.0                      |
| CIFAR100 / Res50     | 10.5         | 6.5          | 7.6           | 8.1           | 40.2             | 11.3                       | 1,872.4                   |
| ImageNet / Res50     | 7.4          | 11.4         | 7.3           | 8.1           | 160.2            | 10.0                       | N/A                       |



                               Table 2: GPU memory (in Gigabytes) during training

| Model                | Standard | Fixed ETF | Implicit ETF w/o DDN Bwd | Implicit ETF w/ DDN Bwd |
|----------------------|:---------:|:---------:|:-------------------------:|:------------------------:|
| UFM-10               | 1.5       | 1.5       | 1.5                       | 1.6                      |
| UFM-100              | 1.7       | 1.7       | 1.7                       | 10.7                     |
| UFM-200              | 1.7       | 1.7       | 1.8                       | 70.3                     |
| UFM-1000             | 1.9       | 1.9       | 2.9                       | N/A                      |
| CIFAR10 / Res18      | 2.2       | 2.2       | 2.2                       | 2.3                      |
| CIFAR10 / Res50      | 2.6       | 2.6       | 2.7                       | 2.8                      |
| CIFAR100 / Res50     | 2.6       | 2.6       | 2.7                       | 18.9                     |
| ImageNet / Res50     | 27.5      | 27.2      | 27.8                      | N/A                      |

The attached PDF includes figures of additional results requested by reviewers, which we discuss in the individual responses below.

---

### Decision · Program_Chairs · 2024-09-25

**Decision:**

Accept (poster)

**Comment:**

The paper exploits the neural collapse phenomenon to improve training convergence and stability. Specifically, by leveraging the duality between penultimate layer features and the final layer weights, along with the simplex ETF structure of the final layer weights, the proposed approach gradually guides the weights towards an optimal simplex ETF during training. This approach differs from existing work that uses a fixed simplex ETF during training, which may require the feature mapping to do the heavy lifting to align features with the classifier. Experimental results on various neural network architectures and real-world datasets demonstrate the improvement of the proposed approach in both training and testing performance.

Overall, all the reviewers found the contribution of this work to be solid and a valuable addition to the exploration of neural collapse for improving training.